# When majority rules, minority loses: bias amplification of gradient descent

**François Bachoc**
University of Lille
Institut Universitaire de France (IUF)
`francois.bachoc@univ-lille.fr`

**Jérôme Bolte**
Toulouse School of Economics
ANITI
`jerome.bolte@tse-fr.eu`

**Ryan Boustany**
Toulouse School of Economics
`ryan.boustany@tse-fr.eu`

**Jean-Michel Loubes**
Université de Toulouse
ANITI & Regalia INRIA
`jean-michel.a.loubes@inria.fr`

## Abstract

Despite growing empirical evidence of bias amplification in machine learning, its theoretical foundations remain poorly understood. We develop a formal framework for majority-minority learning tasks, showing how standard training can favor majority groups and produce stereotypical predictors that neglect minority-specific features. Assuming population and variance imbalance, our analysis reveals three key findings: (i) the close proximity between "full-data" and stereotypical predictors, (ii) the dominance of a region where training the entire model tends to merely learn the majority traits, and (iii) a lower bound on the additional training required. Our results are illustrated through experiments in deep learning for tabular and image classification tasks.

## 1 Introduction

Imbalanced data are pervasive in machine learning, spanning rare-event detection, fraud, faults, medical anomalies, security, finance, and modern LLM pipelines with unequally represented subpopulations, see e.g., [26, 29] for some references. A sensitive case arises in fairness-related applications, where decisions apply to human beings [3, 10, 12, 33]. Addressing this issue is increasingly important notably under regulatory frameworks such as the European Union's AI Act, which emphasizes non-discrimination and risk mitigation.

In all these settings, the goal is to learn predictors that genuinely capture minority structure.

Our focus is on scenarios with two distinctive characteristics. First, the imbalance is typically significative, as we work directly with raw data without resampling or augmentation. Second, the imbalance is not corrected at the data level but addressed only through the training dynamics of gradient descent. An empirical fact, well known to practitioners, is that imbalance is not only preserved but often amplified by training: models initially align with the majority component and only later start to capture minority features. In the fairness literature, this is sometimes referred to as bias amplification [7, 18, 45, 46, 47]. Related simplicity-driven behaviors have been observed in representation learning [6, 16, 22, 39, 44].

This phenomenon has been documented since the 1990s in the class imbalance literature, see [1], which motivated numerous work and heuristics: data-level remedies (oversampling, under-sampling, synthetic examples [9, 43]), algorithm-level adjustments [15], cost-sensitive learning [14], focal and reshaped losses [30, 35]. With the advent of deep learning, the issue became even more acute, as

39th Conference on Neural Information Processing Systems (NeurIPS 2025).

high-capacity models and standard training budgets (a few hundred epochs) tend to privilege majority signals [26]. Despite this history, the mathematical mechanisms of imbalance amplification are still poorly understood geometrically, especially in nonlinear nonconvex regimes [37]. Our goal is to clarify these mechanisms which are essential for sensitive applications.

Using Kantorovich-type arguments, we develop a theoretical and a geometrical framework that explains why and how gradient-based training first produces stereotypical predictors, i.e. aligned merely with the majority, before catching-up and entering a debiasing phase where minority features start to influence predictions.

**Contributions.** They are as follows:
— We first formalize the problem as a generic majority-minority learning task $\min L := L_1 + L_0$, with $L_0 \ll L_1$ using second-order differentiability domination. We prove that each critical point of $L$, which corresponds to a predictor, can be paired with a critical point of $L_1$, termed *stereotypical predictor* (Section 2.2). We bound their distance: it is what we call the *stereotype gap*.
— The proximity of $L$ and $L_1$ implies that the region where minimizing $L$ is 'equivalent' to minimizing $L_1$ occupies nearly the entire parameter space (Section 3.2). This results in a close overlap between $L_1$ training gradient path and the actual training path, illustrating how standard training neglects minority-specific characteristics (Appendix B.2). This proximity between learning curves, as the proximity between representative and stereotypical predictor, is somehow deceptive, since the minority features lie precisely in what differentiates them.
— We prove that gradient descent may require a fairly long training time to merely identify stereotypical predictors, ignoring minority-specific aspects (Section 3.3). A common training failure is when a long initial training phase stalls at a stereotypical predictor. Although this predictor lies close to its corresponding representative predictor, escaping that neighborhood, and thus debiasing the model, often requires much more training because the gradients there are tiny. We derive a lower bound on this extra training duration. The corresponding ratio is called the *catch-up overcost ratio*. It is a debiasing overcost. It quantifies the additional training time required to achieve unbiased predictions.
— We illustrate our theoretical findings through numerical experiments on tabular and image-classification tasks with deep neural networks (Section 4). Minority awareness emerges in preliminary experiments and appears linked to training duration; it also persists under alternative learning strategies as AdamW or XGBoost.

**Related literature.** The bias amplification phenomenon is well documented experimentally, see, for example, [6, 18, 37] and references therein. Yet few theoretical results clarify its causes. The earliest paper we are aware of that addresses the issue is [1] via a diagnostic of an early-phase majority bias and via some algorithmic fix (bisect classwise gradients) with empirical speedups. In a Gaussian setting with ridge regression, [42] show that, for a single pooled model, the between-group gap in expected test risk can exceed the corresponding gap obtained by training separate models for each group. Leveraging the closed form of the ridge estimator, they analyze the asymptotic behavior of this bias-amplification measure. In a related direction, [31] introduce a parametric Gaussian-mixture framework with tunable imbalance and derive analytic ridge-regression solutions, comparing group-wise risks for a jointly trained model versus per-group models. All these results rely on analytic solutions and their asymptotics. Complementing this line, [15] analyze optimization dynamics and articulate theoretical conditions that clarify a phenomenon they term minority initial drop (MID) – an early deterioration of minority recall driven by majority-dominated gradients. They also provide sufficient conditions for monotone per-class loss decrease and show that vanilla (stochastic) gradient descent can be sub-optimal under imbalance. Their perspective focuses on loss trajectories and per-class gradients, rather than the parameter-space geometry and time-to-learn bounds we develop below. In this sense, their results are complementary to ours, and a general, model-agnostic theory of bias amplification in modern ML remains largely open.

**Notations.** Notations on matrices, differential calculus and geometry, that are used throughout the paper, can be found in Appendix A.1.

## 2 Predictions for majority-minority problems in machine learning

We first present our majority-minority scenario in Section 2.1 as a minimization problem: $\min L := L_1 + L_0$. We aim at estimating the distance between a predictor obtained by minimizing the total loss $L$ and a neighboring majority-based predictor obtained by minimizing $L_1$. In practice, the latter may

represent a biased or stereotyped view that a user holds about the underlying problem. We show that a small population and low variance for the minority group lead to proximity between the predictor and the majority-based predictor, making them difficult to distinguish. Our results are first presented for abstract equations (Proposition 6) and general variational problems (Theorem 1); discussions on learning appear in Section 2.3 and in the Appendix.

## 2.1 The setting: majority-minority model and generic losses

**A majority-minority model.** We consider $n$ observations of a variable $Z := (X, Y) \in \mathbb{R}^d \times \mathbb{R}$ ($d > 0$) that can be divided into two groups following the values of a binary variable $A \in \{0, 1\}$. In our scenario, the data are unbalanced: there is a majority group $A = 1$ (with cardinality denoted $n_1$) and a minority group $A = 0$ (cardinality $n_0$), typically with $n_0 \ll n_1$. This heterogeneity, i.e., the variable $A$, may be unknown to the user.

Consider a collection of models or predictors $f_\theta : \mathbb{R}^d \mapsto \mathbb{R}$ indexed by parameters or weights $\theta \in \mathbb{R}^d$ that are learned by minimizing some empirical loss function over the learning set. Given a discrepancy measure $\ell : \mathbb{R}^2 \to \mathbb{R}_+$ we may define the total, majority and minority losses as: for $\theta \in \mathbb{R}^d$,

$$L(\theta) := \frac{1}{n} \sum_{i=1}^{n} \ell(f_\theta(X_i), Y_i) = \underbrace{\frac{1}{n} \sum_{\substack{i=1,\ldots,n \\ A_i=1}} \ell(f_\theta(X_i), Y_i)}_{:=L_1(\theta)} + \underbrace{\frac{1}{n} \sum_{\substack{i=1,\ldots,n \\ A_i=0}} \ell(f_\theta(X_i), Y_i)}_{:=L_0(\theta)}.$$

In the training phase of a learning process, the parameters are often computed through first order methods and thus eventually through vanishing gradients. Assuming both $\ell$ and $f_\theta$ are differentiable, we are thus led to consider equations of the form: $\nabla L(\theta) = 0$, $\nabla L_j(\theta) = 0$, for $j \in \{0, 1\}$. In a strongly imbalanced scenario, $L_0$ may become negligible with respect to $L_1$, so that the equations $\nabla L_1 = 0$ and $\nabla L = 0$ have very close solutions. On the other hand, this proximity does not prevent solutions to the equation $\nabla L_1(\theta) = 0$ from producing biased or stereotyped predictors as they ignore, by definition, the influence of data underlying $L_0$.

The aim of the following sections is to study this phenomenon and provide a set of assumptions for estimating the distance between full-data and stereotypical predictors.

**Generic losses.** For the rest of the article, we adopt a genericity perspective on loss functions by assuming that their critical points are non-degenerated. For $G : \mathbb{R}^d \to \mathbb{R}$ twice differentiable this means that

$$\nabla G(\theta) = 0 \Rightarrow \nabla^2 G(\theta) \text{ is invertible.}$$

In other words, $G$ is a *Morse function*. These functions are generic in the sense that they form an open dense subset in $C^k(\mathbb{R}^d, \mathbb{R})$ for the $C^2$ topology whenever $k \geq 2$, see e.g., [17].

In the machine learning perspective, this is not extremely demanding as, for a fixed $C^2$ function $G$, perturbations of the form $\mathbb{R}^n \ni x \mapsto G_{\gamma, \epsilon}(x) = G(x) + \gamma \|x - \epsilon\|^2$ with $\gamma > 0$ are Morse for almost all couple $(\gamma, \epsilon) \in \mathbb{R}_+ \times \mathbb{R}^n$ – actually it holds true with linear perturbations, see e.g., [40]. This approach aligns with statistical and learning practices, both through ridge regularization (pioneered in [23] whose use in data science is developed for instance in [19], and references therein) and the weight decay approach in deep learning [8].

## 2.2 Perturbation results for critical points of generic losses

Assume $L = L_1 + L_0$ is a general cost. The spirit of the following results is that $L_1$ corresponds to a majority behavior while $L_0$ is attached to minority features, for instance as in the scenario of Section 2.1. In an analytical setting, it translates into a property of the type: $L_0$ is negligible w.r.t $L_1$ (see the assumptions below). We then aim at comparing $\operatorname{crit} L$ and $\operatorname{crit} L_1$; $\operatorname{argmin-loc} L$ and $\operatorname{argmin-loc} L_1$[1]. Note that the theorem below is a general-purpose perturbation result, it is applied in a machine learning setting in the remaining sections.

**Theorem 1** (Strong imbalance and critical points). *Consider two functions $L_1$ and $L_0$ from $\mathbb{R}^d$ to $\mathbb{R}$ that are two times continuously differentiable, and a subset $K \subset \mathbb{R}^d$.*

*Assume that there are strictly positive numbers $\delta, c, M, \tau$ such that*

---

[1]Recall that notations are provided in Appendix A.1.

- *Strong Morse property: For all $\theta \in K$,*

$$\|\nabla L_1(\theta)\| \leq c \implies \rho_{\min}(\nabla^2 L_1(\theta)) \geq \delta, \tag{1}$$

- *Lipschitz regularity: for all $\theta_1, \theta_2 \in K$*

$$\rho_{\max}\left(\nabla^2 L_1(\theta_1) - \nabla^2 L_1(\theta_2)\right) \leq M\|\theta_1 - \theta_2\|, \tag{2}$$

$$\rho_{\max}\left(\nabla^2 L_0(\theta_1) - \nabla^2 L_0(\theta_2)\right) \leq M\|\theta_1 - \theta_2\|, \tag{3}$$

- *Bounds on the 'minority loss':*

$$\sup_{\theta \in K} \|\nabla L_0(\theta)\| \leq \tau, \tag{4}$$

$$\sup_{\theta \in K} \rho_{\max}\left(\nabla^2 L_0(\theta)\right) \leq \tau. \tag{5}$$

*Assume further that*

$$\tau < \min\left\{\frac{c}{2}, \frac{\delta}{8}, \frac{\delta^2}{32M}\right\}, \tag{6}$$

$$\mathrm{dist}_H\left(\mathrm{crit}\, L_1 \cap K, \mathrm{bdry}\, K\right) \geq \frac{6\tau}{\delta}, \quad \mathrm{dist}_H\left(\mathrm{crit}\, L \cap K, \mathrm{bdry}\, K\right) \geq \frac{6\tau}{\delta}. \tag{7}$$

*Then, for each $\widehat{\theta}_1 \in \mathrm{crit}\, L_1 \cap K$ (resp. $\widehat{\theta} \in \mathrm{crit}\, L \cap K$) there exists a unique corresponding $\widehat{\theta} \in \mathrm{crit}\, L \cap K$ (resp. $\widehat{\theta}_1 \in \mathrm{crit}\, L_1 \cap K$) such that*

$$\|\widehat{\theta}_1 - \widehat{\theta}\| \leq \frac{4\tau}{\delta}$$

*and $\widehat{\theta}, \widehat{\theta}_1$ have the same indexes, that is the same number of strictly negative eigenvalues of the Hessian matrices $\nabla^2 L_1(\widehat{\theta}_1)$ and $\nabla^2 L(\widehat{\theta})$.*

**Corollary 1** (Distances between critical and local minimizer sets). *In the context of Theorem 1, if $\mathrm{crit}\, L_1 \cap K$ is non-empty, then $\mathrm{crit}\, L \cap K$ is non-empty and we have*

$$\mathrm{dist}_H\left(\mathrm{crit}\, L_1 \cap K, \mathrm{crit}\, L \cap K\right) \leq \frac{4\tau}{\delta}. \tag{8}$$

*Also, if $\mathrm{argmin}\text{-}\mathrm{loc}\, L_1 \cap K$ is non-empty then $\mathrm{argmin}\text{-}\mathrm{loc}\, L \cap K$ is non-empty and we have*

$$\mathrm{dist}_H\left(\mathrm{argmin}\text{-}\mathrm{loc}\, L_1 \cap K, \mathrm{argmin}\text{-}\mathrm{loc}\, L \cap K\right) \leq \frac{4\tau}{\delta}. \tag{9}$$

*Finally, for each $\theta \in \mathrm{argmin}\text{-}\mathrm{loc}\, L_1 \cap K$, there is $\theta' \in \mathrm{argmin}\text{-}\mathrm{loc}\, L \cap K$ such that the ball $B(\theta', \frac{6\tau}{\delta})$ contains $\theta$, and $L$ is $\delta/8$ strongly convex on this ball.*

**Comments on Theorem 1 and Corollary 1.** Assumptions (2)–(6) are warranted whenever $\nabla L_0$ is $C^1$–small on $K$, i.e., small with respect to the functional semi-norm

$$\|\nabla L_0\|_{1,\infty} := \max\left\{\max_{a=1,\ldots,d} \sup_{\theta \in K}\left|\frac{\partial L_0(\theta)}{\partial \theta_a}\right|, \max_{a,b=1,\ldots,d} \sup_{\theta \in K}\left|\frac{\partial^2 L_0(\theta)}{\partial \theta_a \partial \theta_b}\right|\right\}.$$

Assumption (7) simply means that the critical sets are not too close to the boundary of $K$. If the critical sets lie in a compact set, it suffices to choose $K$ large enough to satisfy the assumption.

A simple reading of Theorem 1 is therefore that when $L$ and $L_1$ are sufficiently close (on $K$), they share the same 'geometry', i.e., they have the same number of local minimizers and, more generally, the same number of critical points for a given index, with, in addition, corresponding points lying at small distance from one another.

Note that [32] establishes results similar to Theorem 1 and Corollary 1, but in a different setting: the comparison of theoretical and empirical risks under i.i.d. random data. In contrast, by relying on Kantorovich's method of proof, we impose no assumptions on the data. An instance of Theorem 1 for the special case of linear regression is provided in Appendix B.1.

## 2.3 A machine learning view: the representative and stereotypical predictions

Let us interpret the above within a learning perspective. Under the premises of Theorem 1, we consider a machine learning model with loss $L : \theta \mapsto L(\theta)$ decomposed into a sum $L = L_1 + L_0$ where $L_1$ and $L_0$ respectively correspond to some majority and minority phenomena.

A critical point of $L$ is called a *representative prediction*, as it takes into account all available data encoded within $L$, i.e. both those in $L_1$ and $L_0$[2]. In the majority-minority model, the critical points of $L_1$ ignore data corresponding to the case when $A = 0$, we thus call them *stereotypical predictions*. The quantity $\mathrm{dist}_H (\mathrm{crit}\, L \cap K, \mathrm{crit}\, L_1 \cap K)$ is called the *stereotype gap*.

Roughly speaking Theorem 1 tells us, in particular, that each representative prediction corresponds to one and only one stereotypical prediction and that these predictions are close whenever the ratio

$$\Delta = \rho_{\max}\left(\nabla^2 L_0(\theta)\right) / \rho_{\min}(\nabla^2 L_1(\theta))$$

is uniformly small. This ratio is the key quantity that governs the stereotype gap.

The result is even more accurate, as Theorem 1 shows that the minimizers of $L$ and $L_1$ actually come by pairs as well, so that the stereotypical and representative predictors obtained in practice are 'dangerously' close in a majority-minority scenario. As we will see through theoretical and numerical experiments, this renders the training phase delicate and potentially biased. Using the well-known fact that gradient descent converges to critical points in the Morse case (see next section and Appendix A.4), we may empirically estimate the stereotypical gaps and the associated 'debiasing training time' in our imbalanced setting (see also the following sections).

Protocol (Table 1 opposite): find a stereotypical predictor $\widehat{\theta}_1$ via the gradient flow $-\nabla L_1$ with Kaiming random initialization. Initialize from this predictor $\widehat{\theta}_1$ and follow the flow of $-\nabla L$, with the guarantee (see Corollary 1) of reaching the corresponding representative predictor $\widehat{\theta}$. Use these values to estimate the gap $\mathrm{dist}_H (\mathrm{crit}\, L, \mathrm{crit}\, L_1)$ via proxies like $\|\widehat{\theta} - \widehat{\theta}_1\|$, and to define a debiasing time from $\widehat{\theta}_1$ to its representative $\widehat{\theta}$ using gradient descent on $L$ with stopping criterion $\|\theta_{k+1} - \widehat{\theta}_1\| \geq 0.99\|\theta_k - \widehat{\theta}_1\|$.

Table 1: Stereotypical and representative predictions for imbalanced CIFAR-2 ($n_0/n \approx 3\%$, see Appendix F.1) with ResNet 18. We report the average and standard deviation over 30 runs.

| Metric | Mean | $\pm$ Std |
|---|---|---|
| Debiasing time | 469 epochs | $\pm$ 9.4 |
| $\|\widehat{\theta} - \widehat{\theta}_1\|$ | 0.6723 | $\pm$ 0.0083 |
| $\|\widehat{\theta} - \widehat{\theta}_1\|_\infty$ | 0.0353 | $\pm$ 0.0047 |
| $\frac{\|\widehat{\theta} - \widehat{\theta}_1\|}{\|\widehat{\theta}\|}$ | 0.00602 | $\pm$ 0.00007 |

# 3 Learning unbalanced data with the gradient method

## 3.1 Gradient descent training

In this section, we study how gradient descent procedures may bias predictions in the sense that a 'careless training' may yield a stereotypical predictor rather than a representative one. Gradient descent training on a $C^2$ loss $L$ is modeled through the ODE (see Appendix A.4 for the representation of ODE curves):

$$\frac{d}{dt}\theta(t) = -\nabla L(\theta(t)) \text{ with } \theta(0) = \theta_{\mathrm{init}} \in \mathbb{R}^d. \tag{10}$$

The ODE solution is called a *training trajectory*. In Appendix B.2, for the special case of linear regression, we also consider the counterpart $\theta_1(t)$ of $\theta(t)$ with $L$ replaced by $L_1$. We show that $\theta_1(t)$ and $\theta(t)$ are close under strong imbalance.

## 3.2 The majority-training and the majority-adverse zones

For $C^2$ smooth losses $L = L_1 + L_0$, the *majority-training zone* is defined by

$$Z_{\mathrm{maj}} = \{\theta \in \mathbb{R}^d : \langle \nabla L(\theta), \nabla L_1(\theta)\rangle > 0\}.$$

---

[2]It would be more natural to reserve that name for local minimizers, as those are generally obtained after training, but we do so for simplicity.

In this region, descending along the gradient of $L$ also decreases $L_1$, and vice versa. In other words, $Z_{\mathrm{maj}}$ is a zone where training $L$ with gradient descent implies training the majority $L_1$. The *majority-adverse* zone is defined as

$$Z_{\mathrm{maj\text{-}adv}} = \{\theta \in \mathbb{R}^d : \langle \nabla L(\theta), \nabla L_1(\theta) \rangle \leq 0\} \text{ so that } \mathbb{R}^d \setminus Z_{\mathrm{maj}} = Z_{\mathrm{maj\text{-}adv}}. \qquad (11)$$

We can similarly consider the minority-training and the minority-adverse zones. One easily sees that, under the Morse assumption, critical points of $L$ or $L_1$ lie in between $Z_{\mathrm{maj}}$ and $Z_{\mathrm{maj\text{-}adv}}$, (see Proposition 7 in Appendix C.2 for details). In other words, the stereotypical and representative predictors lie on the boundary of $Z_{\mathrm{maj}}$.

We now establish two major facts: first, the majority zone is typically large, meaning that training the entire model often results in learning only the majority traits (see also the illustration of Figure 1); second, the majority-adverse zone promotes the training of the minority loss.

**Theorem 2** (Majority adverse zone). *Let* $K_{-\frac{2\tau}{\delta}} = \{\theta \in K; \mathrm{dist}\,(\theta, \mathrm{bdry}\,K) \geq \frac{2\tau}{\delta}\}$. *Under Theorem 1 assumptions:*

$$Z_{\mathrm{maj\text{-}adv}} \cap K_{-\frac{2\tau}{\delta}} \subset \bigcup_{\widehat{\theta}_1 \in \mathrm{crit}\,L_1 \cap K} B\left(\widehat{\theta}_1, \frac{2\tau}{\delta}\right).$$

$$\subset \bigcup_{\widehat{\theta}_1 \in \mathrm{crit}\,L_1 \cap K} B\left(\widehat{\theta}_1, \frac{1}{4}\right).$$

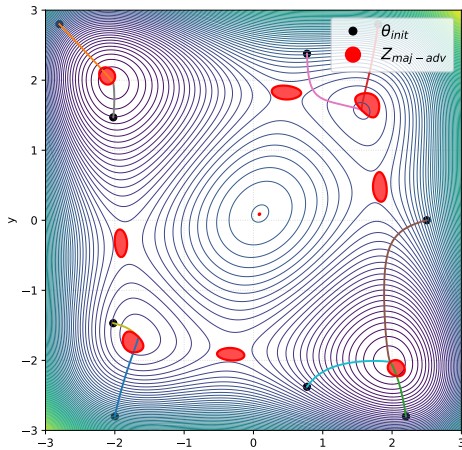

**Remark 1** (On the majority-training zone size). Under the assumptions of Theorem 1, in the high dimensional regime the majority adverse zone has a volume lower than $O(4^{-d})$ —much lower in general as we have chosen a conservative bound. Note also that the stronger the imbalance, the more negligible it becomes, see the comments after Theorem 1.

**Lemma 1** (The majority adverse zone favors minority). *For* $\theta \in Z_{\mathrm{maj\text{-}adv}}$, *we have*

$$\langle \nabla L(\theta), \nabla L_0(\theta) \rangle \geq 0.$$

*Thus a training trajectory* $\theta : I \to \mathbb{R}^d$ *evolving within* $Z_{\mathrm{maj\text{-}adv}}$ *is such that* $L_0(\theta(t))$ *is non-increasing over the interval* $I$.

In other words, when the trajectory evolves within the majority-adverse zone, the dynamics learns minority features.

Figure 1: The majority region (white) covers nearly the entire space, while the majority-adverse region (red) is small. With random initialization, training typically begins in white; hence majority features are learned during the initial phase, which accounts for most of the path length (though not the time). The gradient trajectory then enters a red zone, where minority features are improved. Despite the short arc length in red, the time spent there may be very long. These transient passages through red before convergence correspond to 'unlucky' curves.

### 3.3 Lower bounds for debiasing duration and catch-up overcost

Although we cannot, at this stage, provide worst–case 'biasing' complexity bounds, we can obtain a *lower bound* by placing ourselves in a setting with a high risk of bias toward the majority. Consider an 'unlucky gradient curve' $t \mapsto \theta(t)$ solving (10) that effectively ignores the minority until it meets a majority predictor. On $[0, t_{\mathrm{stereotype}}]$, the trajectory carries the initial condition $\theta_{\mathrm{init}}$ to a critical point of $L_1$, viewed as a stereotype and denoted

$$\widehat{\theta}_{\mathrm{stereotype}} := \theta(t_{\mathrm{stereotype}}).$$

Up to time $t_{\mathrm{stereotype}}$, it is 'as if' only $L_1$ were trained —the minority is entirely ignored. Thereafter, $\theta(t)$ moves toward a critical point of the full loss $L$, denoted $\widehat{\theta}$, which we interpret as a representative

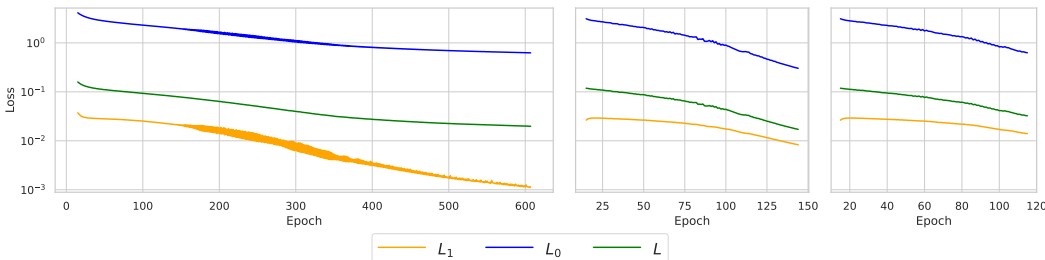

Figure 2: Training curves on 'Imbalanced CIFAR 2' with ResNet18 (see Appendix A.2). Left to right: unlucky curve with stopping rule based on minority recognition, i.e., $\text{Acc}_0 > 99\%$; random trajectory with the same rule; random trajectory with global accuracy stopping rule $\text{Acc} > 99\%$. Unlucky initialization has 600 epochs while 'careless training' (third one) needs 100 epochs and has much higher final $L_0$ value. Middle: random initialization with minority aware stopping rule training has 140 epochs. In the real world $\text{Acc}_0 > 99\%$ is not a realistic criterion as we do not know the minority class. Conclusion: risk-averse training should rely on considerably longer training (here +500%), confirming the results of Section 3.3. For more confident training, substantially longer runs are still required (here +40%).

predictor. By the Cauchy-Lipschitz existence theorem, this trajectory typically exists. This curve and neighboring ones may be quite detrimental to fair predictions as shown in Figure 2.

The next proposition shows that $t_{\text{stereotype}}$ is typically large as $\|\widehat{\theta}_{\text{stereotype}} - \widehat{\theta}\|$ is typically very small (see Theorem 1 and Table 2).

**Proposition 1** (Training duration). *Assume that $L$ is twice continuously differentiable on $\mathbb{R}^d$. Consider a ball $\mathcal{B}$ containing $\widehat{\theta}$ and $\{\theta(t); t \geq 0\}$. Assume that for some $M < \infty$ and for all $\theta \in \mathcal{B}$,*

$$\rho_{\max}\left(\nabla^2 L(\theta)\right) \leq M. \tag{12}$$

*Assume $\widehat{\theta} \neq \widehat{\theta}_{\text{stereotype}}$, then $t_{\text{stereotype}} \geq \dfrac{1}{M} \log\left(\dfrac{\|\theta_{\text{init}} - \widehat{\theta}\|}{\|\widehat{\theta}_{\text{stereotype}} - \widehat{\theta}\|}\right)$.*

Let us give a simple yet illustrative example showing that the bound is tight and that training duration becomes rather long in the small step-size regime typical of large-scale deep learning problems.

**Example 1.** (a) (The bound is tight). Consider the elementary but instructive model $L_1(x) = x^2/2$, $L_0 = [\delta(x-c)^2]/2$ with $c, \delta > 0$, $\delta$ being a small imbalance factor that reflects the minority scenario. Simple computations give the representative predictor $\widehat{x} = \frac{\delta}{1+\delta}c$ while the stereotypical predictor is $\widehat{x}_{\text{stereotype}} = 0$. The time to reach the stereotype 0 from $x_{\text{init}} < 0$ is

$$t_{\text{stereotype}} = \frac{1}{1+\delta} \log\left(\frac{\widehat{x} - x_{\text{init}}}{\widehat{x} - \widehat{x}_{\text{stereotype}}}\right) \quad \text{whence Proposition 1 is tight.}$$

(b) (Small steps yield long training duration). Consider now $x_{k+1} = x_k - \eta \nabla L(x_k)$ with a short step $\eta = 10^{-2}$, as it could be done in deep learning. Let $x_{\text{init}} = -2c$; it is a multiple of $c$ for convenience, while remoteness from 0 reflects the ignorance of a blind user on the exact location of the minimizer. Since $|x_{\text{init}} - \widehat{x}| = 2c$ and $|\widehat{x}_{\text{stereotype}} - \widehat{x}| \approx \delta c$:

$$t_{\text{stereotype}} = \frac{1}{1+\delta} \log\left(\frac{|x_{\text{init}} - \widehat{x}|}{|\widehat{x}_{\text{stereotype}} - \widehat{x}|}\right) \approx \frac{1}{1+\delta} \log\left(\frac{2}{\delta}\right).$$

The discrete time when the stereotype is reached may be approximated by $k_{\text{stereotype}} \approx t_{\text{stereotype}}/\eta$. We may provide a table for $\eta = 10^{-2}$, $x_{\text{init}} = -2c$.

| $\delta$ | $t_{\text{stereotype}}$ (bound) | $k_{\text{stereotype}}$ (bound) |
|---|---|---|
| $10^{-2}$ | $\frac{1}{1.01} \log(200) \approx 5.25$ | $\approx 525$ |
| $10^{-3}$ | $\frac{1}{1.001} \log(2000) \approx 7.59$ | $\approx 759$ |
| $10^{-4}$ | $\frac{1}{1.0001} \log(20000) \approx 9.89$ | $\approx 989$ |

Thus strong imbalance together with traditionally cautious DL steps give long training durations.

Next, we provide a lower bound on the extra-time $t_{\text{catchup},\epsilon} - t_{\text{stereotype}}$ needed to achieve the relative $\epsilon$ precision, where $\epsilon \in (0,1)$, $t_{\text{catchup},\epsilon} > t_{\text{stereotype}}$ and

$$\frac{\|\theta(t_{\text{catchup},\epsilon}) - \widehat{\theta}\|}{\|\widehat{\theta}_{\text{stereotype}} - \widehat{\theta}\|} \leq \epsilon. \tag{13}$$

This extra time is interpreted as a catch-up time for the algorithm to detect the minority with an acceptable precision. Indeed from time $t_{\text{stereotype}}$ to $t_{\text{catchup},\epsilon}$, the trajectory $\theta(t)$ leaves 'a stereotype' and becomes closer to 'a representative predictor'. *It is a debiasing phase in which the algorithm progressively removes the bias it has itself created during the preliminary training phase.*

**Proposition 2** (Debiasing duration[3]). *Assume that $L$ is twice continuously differentiable and satisfies* (12), *for the same $M$ and $\mathcal{B}$. Assume that $\widehat{\theta} \neq \widehat{\theta}_{\text{stereotype}}$. For $0 < \epsilon < 1$, consider $t_{\text{catchup},\epsilon}$ such that* (13) *holds. Then* $t_{\text{catchup},\epsilon} - t_{\text{stereotype}} \geq \frac{1}{M} \log\left(\frac{1}{\epsilon}\right)$.

**Example 1** (continued). (On the length of debiasing duration) Back to the setting of Example 1.Again from simple computations, the debiasing time needed to go from the stereotype $\widehat{x}_{\text{stereotype}} = 0$ to a relative precision $\varepsilon \in (0,1)$ around the representative $\widehat{x} = \frac{\delta}{1+\delta} c$ is

$$t_{\text{catchup},\epsilon} - t_{\text{stereotype}} = \frac{1}{1+\delta} \log\left(\frac{1}{\varepsilon}\right),$$

so Proposition 2 is tight. For gradient descent $x_{k+1} = x_k - \eta \nabla L(x_k)$, the error decays geometrically with factor $1 - \eta(1+\delta)$. Thus the number of iterations to reach the same relative precision $\varepsilon$ satisfies

$$k_{\text{catchup}}(\eta, \delta, \varepsilon) \geq \frac{\log(1/\varepsilon)}{-\log(1 - \eta(1+\delta))} \approx \frac{1}{\eta(1+\delta)} \log\left(\frac{1}{\varepsilon}\right),$$

which is a version of Proposition 2. Thus an approximate debiasing step count with a 'standard' ML learning rate $\eta = 10^{-2}$:

| $\delta$ | $\varepsilon = 10^{-1}$ | $10^{-2}$ | $10^{-3}$ |
|---|---|---|---|
| $10^{-2}$ | 228 | 456 | 684 |
| $10^{-3}$ | 230 | 460 | 690 |

Hence, for strong imbalance ($\delta \ll 1$) and small steps, debiasing typically costs a few hundred additional iterations even after reaching the stereotype.

## 4 Numerical experiments

We study the effect of subgroup imbalance in supervised deep learning using image (CIFAR-10 [28], EuroSAT [21]) and tabular (Adult [5]) datasets. Each dataset is denoted by $\mathcal{D} = \{(X_i, Y_i, A_i)\}_{i=1}^{n}$, where $(X_i, Y_i)$ is an input-label pair and $A_i \in \{0,1\}$ is a binary attribute (0 is minority). While $A$ is not used during training, it enables evaluation of model performance across imbalanced subgroups. In each experiment, we report the global loss $L = L_0 + L_1$, and *average loss per sample* in each group, i.e., $(nL_0)/n_0$ and $(nL_1)/n_1$. For details on the implementation setup, see Appendix F.

**Metrics for class-balanced predictions.** We evaluate class-balance using training (and occasionally test) accuracy, denoted by $\text{Acc}$, $\text{Acc}_0$ (see Appendix A.2), as our focus is on optimization under imbalance. To measure the time cost of 'well balanced training', we track the number of epochs $t$ needed to reach a threshold accuracy level $\kappa \in [0,1]$ :

$$T_{\text{early}} := \min_{t \in \mathbb{N}}\{\text{Acc}(\theta_t) \geq \kappa\}, \quad T_{\text{final}} := \min_{t \in \mathbb{N}}\{\text{Acc}_0(\theta_t) \geq \kappa\}, \quad T_{\text{debias}} := T_{\text{final}} - T_{\text{early}}.$$

We define the *catch-up overcost* as the relative delay to reach good class-balance prediction, i.e., a satisfying minority accuracy:

$$\text{Catch-up Overcost} := \frac{T_{\text{debias}}}{T_{\text{early}}}.$$

---

[3]See also Proposition 4 in Appendix B.3 for complementary results on relative values of $L_0$.

**Imbalanced CIFAR-10.** We investigate the effect of class imbalance on CIFAR-10 using models from 100K to 25M parameters (see Table 2). The original dataset has 10 classes with 5000 samples each. To create imbalance, we subsample one class (denoted $A = 0$) to retain $n_0$ samples, and keep the others ($A = 1$) unchanged with $n_1 = 9 \times 5000$. As in [26], we define the imbalance ratio as $\zeta = n_0/5000$, which gives a group proportion $n_0/(n_0 + n_1) = \zeta/(\zeta + 9)$, and consider four imbalance levels: $\zeta \in \{1\%, 10\%, 30\%, 80\%\}$. In Figure 3, we show the results for ResNet-18 (see also Appendix D for more). For $\zeta = 1\%$, $Acc_0$ remains close to zero for about 60 epochs, following a stereotypical training curve (see Appendix).

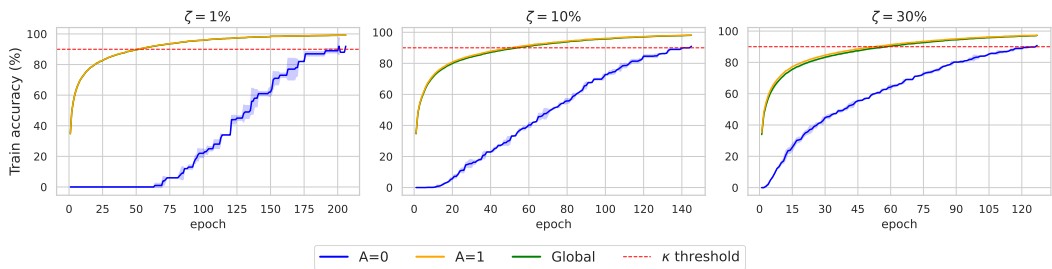

Figure 3: Training accuracy for different subgroup imbalance scenarios (1%, 10%, and 30%) using ResNet18 on CIFAR-10 and threshold $\kappa = 90\%$. Greater imbalance delays the learning of minority features: their accuracy reaches $\kappa$ later.

Table 2: Catch-up overcost (in %) for each model across imbalance levels $\zeta \in \{1\%, 10\%, 30\%, 80\%\}$. We report means over 3 runs with thresholds $\kappa \in \{90\%, 99\%\}$, and model parameter counts.

| Models | Number of parameters | $\kappa = 90\%$ | | | | $\kappa = 99\%$ | | | |
|---|---|---|---|---|---|---|---|---|---|
| | | 1% | 10% | 30% | 80% | 1% | 10% | 30% | 80% |
| MobileNetV2 [38] | 543K | 450 | 275 | 166 | 0 | 62 | 52 | 42 | 0 |
| SqueezeNet [25] | 727K | 270 | 203 | 150 | 0 | 55 | 53 | 32 | 0 |
| VGG11 [41] | 9M | 291 | 171 | 114 | 0 | 53 | 44 | 25 | 0 |
| ResNet18 [20] | 11M | 292 | 164 | 113 | 0 | 61 | 49 | 31 | 0 |
| VGG19 | 20M | 280 | 152 | 112 | 0 | 70 | 65 | 50 | 0 |
| ResNet50 | 25M | 157 | 86 | 68 | 0 | 50 | 37 | 37 | 0 |
| ResNet101 | 42M | 145 | 90 | 64 | 0 | 34 | 36 | 26 | 0 |

**EuroSAT.** We use a ResNet18 model and evaluate its behavior on a binary classification task derived from the EuroSAT dataset. Images are labeled according to a binary attribute $A$, where $A = 0$ corresponds to bluish images ($n_0/(n_0 + n_1) \approx 0.03$) and $A = 1$ to all others. We do not modify the class proportions and use the imbalance present in the original dataset. Figure 4 displays losses and accuracies for both subgroups evidencing a catch-up overcost of 45% for a threshold $\kappa = 90\%$.

**Adult income census.** We train a TabNet classifier [2] on a binary task from the Adult dataset [5]. The minority group ($A = 0$) includes high-income women, representing only 3% of the training set. The majority group ($A = 1$) includes all others. We preserve the original class distribution and train with cross-entropy loss, tracking subgroup metrics. Results are shown in Figure 5 and we have a catch-up overcost of 416% for a threshold $\kappa = 90\%$.

**Results and discussion.** Fairness under imbalance requires much longer training: the minority group ($A = 0$) consistently reaches $\kappa$ much later than the global accuracy. The catch-up overcost is particularly high under strong imbalance, exceeding 400% on Adult and CIFAR-10. Empirically, larger models reduce this overcost but do not eliminate the necessity of longer well-tailored training. These results support our theoretical findings on debiasing duration in imbalanced settings (see Section 3.3), the overwhelming dominance of the majority-training zone, and the difficulty of

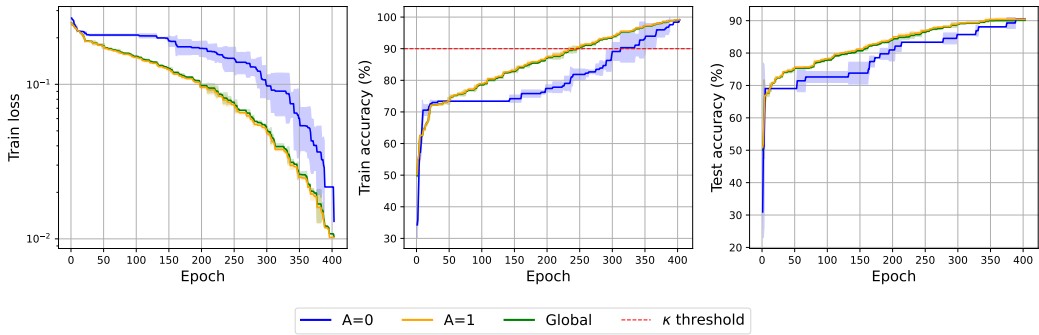

Figure 4: Training and test loss/accuracy for ResNet-18 on EuroSAT (mean of 3 runs). Minority classes exhibit delayed learning – their accuracy improves substantially only in later epochs.

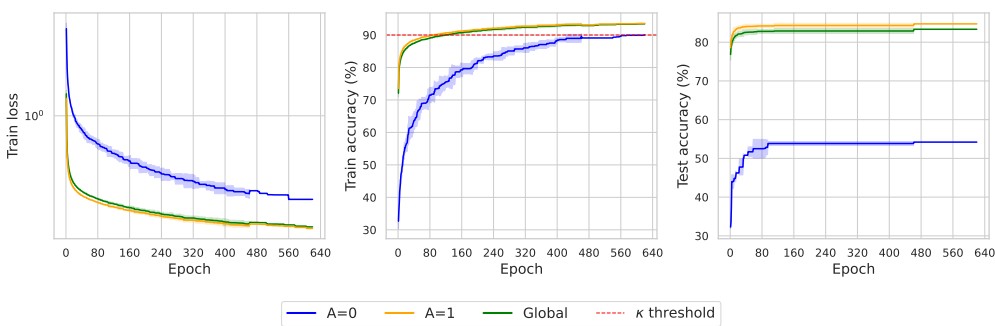

Figure 5: Loss and accuracy with TabNet on Adult (mean of 3 runs). Under strong imbalance, the catch-up overcost is substantial — around 400%.

distinguishing a representative predictor from a stereotypical one. We also ran preliminary experiments with AdamW and XGBoost (gradient-boosted decision trees). In both cases, we observe the same qualitative phenomenon. Irrespective of faster or slower absolute training, attaining minority awareness requires a comparable relative increase in training: extra epochs/steps for AdamW and extra boosting rounds for XGBoost; see Appendix D.1 and Appendix E.

## 5   Conclusion

Although our goals are primarily theoretical and future research should explore more refined training protocols, we can draw several conclusions supported by both theory and numerics —ours and the community's as well, see e.g., [1, 29]. These conclusions may also serve as recommendations for practitioners. Two key quantities emerge as critical in our study: the stereotype gap and the training duration. Additionally, we have empirical evidence that the model size may be a determining factor in achieving budget frugality.

— In a majority-minority scenario, population and variability imbalance are determining factors influencing the stereotype gap (Theorem 1 and the subsequent subsections). This gap, between stereotypes and representative predictors, can be very small in severely imbalanced cases.

— For convex or deep learning problems, gradient training generally leads to a 'satisfying predictor' in the sense of a low-value loss $L$, see e.g., [4] or [13]. However, in our majority-minority scenario, the action of $L_0$ is generally almost indetectable, as shown in Figure 1 and Section 3.3, thus early stopping and under-dimensioned models are prone to produce stereotypes.

— To obtain a representative predictor, it is advisable to use larger networks and extend the training duration, as supported by Propositions 1 and 2, and the numerical section. The corresponding catch-up overcost ratio can take considerable values, e.g., from $25\%$ to $450\%$ for the imbalanced CIFAR-10. However, this must be mitigated in view of possible spurious correlations that arise in overparameterized regimes [37].

## Acknowledgments and Disclosure of Funding

The authors are grateful for the feedback by the anonymous reviewers, that led to considerable improvement of the paper. This work was supported by the ANR project Regul IA and by the Chairs TRIAL and UQPhysAI of the Toulouse ANITI AI Cluster. JB acknowledges support from the Air Force Office of Scientific Research, Air Force Material Command, USAF, under grant number FA8655-22-1-7012 and TSE-P. Access to MesoNET resources in Toulouse was granted under allocation m23038.

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

## Contents

## A  Notations and auxiliary results

### A.1  Notations.

For a matrix $A$, we write $\rho_{\min}(A)$ and $\rho_{\max}(A)$ for its smallest and largest singular value. If the matrix $A$ is square symmetric, we write $\lambda_{\min}(A)$ and $\lambda_{\max}(A)$ for its smallest and largest eigenvalue. Given $x \in \mathbb{R}^d$, we write $\|x\|$ for its Euclidean norm and for $\epsilon > 0$, we write $B(x, \epsilon) = \{y \in \mathbb{R}^d; \|x - y\| \leq \epsilon\}$.

For a function $f : \mathbb{R}^d \to \mathbb{R}^d$ and for $x \in \mathbb{R}^d$, we write $\operatorname{Jac} f(x)$ for the Jacobian matrix of $f$ at $x$. For a function $f : \mathbb{R}^d \to \mathbb{R}$ and for $x \in \mathbb{R}^d$, we write $\nabla f(x)$ for the gradient vector of $f$ at $x$ and $\nabla^2 f(x)$ for the Hessian matrix of $f$ at $x$. For a function $G : \mathbb{R}^d \to \mathbb{R}$, we write

$$\operatorname{crit} G = \{\theta \in \mathbb{R}^d : \nabla G(\theta) = 0\}$$
$$\operatorname{argmin-loc} G = \{\theta \in \mathbb{R}^d : \theta \text{ is a local minimizer of } G \text{ over } \mathbb{R}^d\}.$$

For a non-empty subset $A$ of $\mathbb{R}^d$ and for $x \in \mathbb{R}^d$, we let $\operatorname{dist}(x, A) = \inf_{y \in A} \|x - y\|$. For two non-empty subsets $A$ and $B$ of $\mathbb{R}^d$, the Hausdorff distance between $A$ and $B$ is denoted by

$$\operatorname{dist}_{\mathrm{H}}(A, B) = \max\left(\sup_{x \in A} \inf_{y \in B} \|x - y\|, \sup_{y \in B} \inf_{x \in A} \|x - y\|\right).$$

When $A$ and $B$ are non-empty and bounded this quantity is finite.

The topological boundary of $A$ is written $\operatorname{bdry} A$.

### A.2  Training metrics

Let $f_\theta : \mathbb{R}^d \to \mathbb{R}^C$ be a neural network, parameterized by $\theta$, that maps an input $x \in \mathbb{R}^d$ to a vector of $C$ class scores. We denote $[C] = \{1, \ldots, C\}$ the set of class indices, and define the predicted label as $\hat{y}(x) = \arg\max_{c \in [C]} f_\theta(x)_c$. We compute accuracy separately for each group $j \in \{0, 1\}$ as the proportion of correct predictions in $\mathcal{D}_{A=j}$, and define the *global accuracy* as the weighted average across groups. For each $j \in \{0, 1\}$:

$$\operatorname{Acc}_j(\theta) = \frac{1}{n_j} \sum_{(x_i, y_i) \in \mathcal{D}_{A=j}} \mathbb{1}[\hat{y}(x_i) = y_i], \text{ and } \operatorname{Acc}(\theta) = \frac{n_0}{n} \operatorname{Acc}_0(\theta) + \frac{n_1}{n} \operatorname{Acc}_1(\theta),$$

where $\mathbb{1}[\hat{y}(x_i) = y_i]$ denotes the indicator function, equal to 1 if the predicted label matches the true label and 0 otherwise.

### A.3 Lemma

The next lemma is well-known but stated here for convenience.

**Lemma 2.** *Let $E$ be an open set of $\mathbb{R}^k$ for some $k \in \mathbb{N}$. Let $f : E \to \mathbb{R}^k$ have Jacobian $\mathrm{Jac}\, f$. Let $x, y \in E$ so that the segment between $x$ and $y$ is in $E$. Then*

$$\|f(y) - f(x)\| \leq \left( \sup_{u \in E} \rho_{\max}(\mathrm{Jac}\, f(u)) \right) \|y - x\|.$$

### A.4 Discretization of ODE curves

In various parts of this paper, we refer to or represent ODE curves in our experiments. Unless otherwise specified, this refers to a discretization of the ODE using small step sizes. For instance, given the dynamics

$$\dot{\theta}(t) = F(\theta(t)), \quad \theta(0) = \theta_{\mathrm{init}},$$

with $F : \mathbb{R}^p \to \mathbb{R}^p$ a locally Lipschitz field, the discretization we use is of the form

$$\theta_{k+1} = \theta_k - s_k F(\theta_k),$$

where the step size $s_k \ll 1$; in practice, we typically use $s_k = O(10^{-3})$.

Note however that for the numerical section, we proceed differently as our objective is rather training through the gradient method. We thus use larger steps and mini-batches.

## B  A case study: linear regression

To illustrate further our results in Sections 2 and 3, consider a multidimensional regression model with loss

$$L(\theta) = \frac{1}{2n} \|X\theta - Y\|^2 = \underbrace{\frac{1}{2n} \|X^1\theta - Y^1\|^2}_{L_1(\theta)} + \underbrace{\frac{1}{2n} \|X^0\theta - Y^0\|^2}_{L_0(\theta)}, \tag{14}$$

where $X$ is $n \times d$ with rows $X_1^\top, \ldots, X_n^\top$, and where $X^1$ (respectively $X^0$) contains the rows of $X$ from the majority (respectively minority) class. Similarly, $Y$ is $n$-dimensional with components $Y_1, \ldots, Y_n$ and $Y^1$ (respectively $Y^0$) contains the components of $Y$ from the majority (respectively minority) class. Letting $X^{j\top} = (X^j)^\top$ for $j = 0, 1$, we define the corresponding empirical covariance matrices $S = X^\top X/n$, $S_0 = X^{0\top} X^0/n_0$, $S_1 = X^{1\top} X^1/n_1$. Assume that the covariance matrices are invertible, which may be granted through a ridge regression model in the generic/regularized spirit presented in Section 2.1.

### B.1 Distance between minimizers

In the setting of linear regression, Theorem 1 in Section 2 becomes the following (simpler) theorem.

**Theorem 3** (Representative-stereotypical gap: linear regression case). *Assume that $S_0$ and $S_1$ are invertible, and denote by $\widehat{\theta}$, $\widehat{\theta}_1$, and $\widehat{\theta}_0$, respectively, the unique global minimizers of $L$, $L_1$, and $L_0$, respectively (on $\mathbb{R}^d$). Then*

$$\|\widehat{\theta} - \widehat{\theta}_1\| \leq \frac{2\rho_{\max}(n_0 S_0)}{\rho_{\min}(n_1 S_1)} \left( 1 + \|\widehat{\theta}_1 - \widehat{\theta}_0\| \right).$$

The key quantity behind the stereotypical gap is the ratio

$$\frac{\rho_{\max}(n_0 S_0)}{\rho_{\min}(n_1 S_1)} = \frac{\rho_{\max}(\nabla^2 L_0)}{\rho_{\min}(\nabla^2 L_1)},$$

where we have omitted the dependence on $\theta$ in the Hessians, which are constant. Two statistical effects drive this ratio:

— Population size ratio: when the majority is much larger than the minority, then $n_0/n_1$ is small, this tends to increase the risk of stereotypical predictions.

— Min-max variability ratio: if the smallest variability of the majority is much bigger than the largest variability of the minority group then stereotypical predictions are more likely.

## B.2 Stereotypical and representative training curves

In this section, we compare training $L$ as in (10), which provides a *representative training curve*, with training on the majority group, which provides a *stereotypical training curve* as if the minority did not exist. The stereotypical training curve is:

$$\frac{d}{dt}\theta_1(t) = -\nabla L_1(\theta_1(t)).$$

Our estimate depends once more on the ratio $\Delta = \rho_{\max}(n_0 S_0)/\rho_{\min}(n_1 S_1)$.

**Proposition 3** (Distance between stereotypical and representative training curves)**.** *We have, for any $t > 0$,*

$$\|\theta(t) - \theta_1(t)\| \le \|\widehat{\theta} - \widehat{\theta}_1\| + t\rho_{\max}\left(\frac{n_0}{n}S_0\right)e^{-t\rho_{\min}\left(\frac{n_1}{n}S_1\right)}\left(\|\widehat{\theta}_1\| + \|\theta_{\mathrm{init}}\|\right),$$

$$\|\theta - \theta_1\|_\infty := \sup_{t>0}\|\theta(t) - \theta_1(t)\| \le \|\widehat{\theta} - \widehat{\theta}_1\| + \frac{\|\widehat{\theta}_1\| + \|\theta_{\mathrm{init}}\|}{e}\frac{\rho_{\max}(n_0 S_0)}{\rho_{\min}(n_1 S_1)}.$$

## B.3 Catch-up overcost as measured with the minority loss $L_0$

Proposition 2 measures the catch-up overcost as the time needed to get close to $\widehat{\theta}$ as measured by the distance between parameters. The following proposition shows more qualitatively, for the linear model, that the catch-up overcost goes to infinity, when it is defined as the time needed to get close to $\widehat{\theta}$ as measured by the minority loss $L_0$.

**Proposition 4** (Catch-up overcost measured with the loss $L_0$)**.** *Assume that $\widehat{\theta}$, $\widehat{\theta}_0$ and $\widehat{\theta}_1$ are two-by-two distinct. Consider a representative training curve $t \mapsto \theta(t)$ as in (10), such that for some $t_{\mathrm{stereotype}}$, $\theta(t_{\mathrm{stereotype}}) = \widehat{\theta}_1$. Assume that for $t \ge t_{\mathrm{stereotype}}$, $\theta(t) \in Z_{\mathrm{maj\text{-}adv}}$.*

*For $0 < \epsilon < 1$, consider $t'_{\mathrm{catchup},\epsilon}$ such that*

$$\frac{L_0(\theta(t'_{\mathrm{catchup},\epsilon})) - L_0(\widehat{\theta})}{L_0(\widehat{\theta}_1) - L_0(\widehat{\theta})} \le \epsilon. \tag{15}$$

*Then we have*

$$t'_{\mathrm{catchup},\epsilon} - t_{\mathrm{stereotype}} \xrightarrow[\epsilon \to 0]{} \infty.$$

## B.4 The minority-adverse zone can be large

The minority-adverse zone is defined as

$$Z_{\mathrm{min\text{-}adv}} = \{\theta \in \mathbb{R}^d : \langle \nabla L(\theta), \nabla L_0(\theta)\rangle \le 0\}$$

and is the counterpart to the majority-adverse zone in (11). The next proposition exhibits a ball of radius $R$ that is contained in the minority-adverse zone. This radius $R$ is large whenever $\|\widehat{\theta} - \widehat{\theta}_0\|$ is large and $S$ and $S_0$ are well-conditioned. Hence, roughly speaking, while Theorem 2 states that the majority-adverse zone is always small, the next proposition states that the minority-adverse zone can be large. Hence, gradient descents on $L$ may not decrease $L_0$ over long training times, which is a conclusion of our numerical experiments in Section 4.

**Proposition 5.** *Assume $\widehat{\theta} \ne \widehat{\theta}_0$. Let*

$$R = \frac{\rho_{\min}(S_0)\rho_{\min}(S)}{33\rho_{\max}(S_0)\rho_{\max}(S)}\|\widehat{\theta} - \widehat{\theta}_0\|. \tag{16}$$

*Then there exists $\overline{\theta} \in \mathbb{R}^d$ such that $B(\overline{\theta}, R) \subset Z_{\mathrm{min\text{-}adv}}$.*

# C Proofs and extra results

## C.1 Proofs and extra results of Section 2.2

**Kantorovich theorem.** A great part of Section 2.2 relies on a theorem of Kantorovich type for Newton's method [11, Theorem 5] whose proof is based on [27]. This result is recalled below:

**Theorem 4** (Newton–Kantorovich Theorem 'with only one constant' (existence))**.** *Let $\theta^\star \in \mathbb{R}^d$ and $\widetilde{R} > 0$. Let $\Omega$ be an open set containing the closed ball $B(\theta^\star, \widetilde{R})$. Let $G : \Omega \to \mathbb{R}^d$ be a continuously differentiable mapping. Suppose that the following conditions are satisfied:*

(K1) $\mathrm{Jac}\, G(\theta^\star)$ *is invertible with* $\|\mathrm{Jac}\, G(\theta^\star)^{-1} G(\theta^\star)\| \leq \frac{\widetilde{R}}{2}$.

(K2) *For all* $\theta, \theta' \in B(\theta^\star, \widetilde{R})$, $\rho_{\max}\left( \mathrm{Jac}\, G(\theta^\star)^{-1} \left( \mathrm{Jac}\, G(\theta) - \mathrm{Jac}\, G(\theta') \right) \right) \leq \frac{\|\theta - \theta'\|}{\widetilde{R}}$.

*Then there exists a unique $\widetilde{\theta} \in B(\theta^\star, \widetilde{R})$ such that $G(\widetilde{\theta}) = 0$.*

We need beforehand abstract results on equation perturbations. Let $\theta^\star \in \mathbb{R}^d$. We consider a function $F : \mathbb{R}^d \to \mathbb{R}^d$ such that $F(\theta^\star) = 0$. Let $p : \mathbb{R}^d \to \mathbb{R}^d$ and consider the equation defined for $\theta \in \mathbb{R}^d$,

$$F(\theta) = p(\theta). \tag{17}$$

If the function $p$ is negligible, in a certain sense, with respect to the dominant term $F(\theta)$, (17) becomes a perturbed version of equation $F(\theta) = 0$. Its solution will be close to the solution of the non perturbed equation, $\theta^\star$. Proposition 6 quantifies partly this phenomenon.

**Proposition 6** (Distance to a perturbed solution)**.** *Assume $F$ and $p$ are continuously differentiable and that there are strictly positive numbers $\delta, M, \tau$ such that:*

- *Conditioning of $F$ and $F - p$*

$$\rho_{\min}(\mathrm{Jac}\, F(\theta^\star)) \geq \delta \quad and \quad \rho_{\min}(\mathrm{Jac}\,(F - p)(\theta^\star)) \geq \delta, \tag{18}$$

- *Differential regularity of the nonlinear equation*

$$\rho_{\max}\left( \mathrm{Jac}\, F(\theta) - \mathrm{Jac}\, F(\theta') \right) \leq M\|\theta - \theta'\|, \qquad \theta, \theta' \in B\left( \theta^\star, \frac{2\tau}{\delta} \right), \tag{19}$$

$$\rho_{\max}\left( \mathrm{Jac}\, p(\theta) - \mathrm{Jac}\, p(\theta') \right) \leq M\|\theta - \theta'\|, \qquad \theta, \theta' \in B\left( \theta^\star, \frac{2\tau}{\delta} \right), \tag{20}$$

- *Perturbation bounds*

$$\|p(\theta^\star)\| \leq \tau, \tag{21}$$

$$\rho_{\max}(\mathrm{Jac}\, p(\theta^\star)) \leq \tau. \tag{22}$$

*If the perturbation ratio $\tau/\delta$ satisfies*

$$\tau/\delta < \frac{\delta}{4M}, \tag{23}$$

*then, there is a unique $\theta_p$ solution to $F(\theta_p) = p(\theta_p)$, which is close to the solution of $F(\theta^\star) = 0$, in the sense that*

$$\theta_p \in B\left( \theta^\star, \frac{2\tau}{\delta} \right).$$

*Proof of Proposition 6.* For $\theta \in \mathbb{R}^d$, let $G(\theta) = F(\theta) - p(\theta)$. We apply Kantorovich's Theorem above (Theorem 4) to the function $G$. The quantity $\widetilde{R}$ is taken as

$$\widetilde{R} = \frac{2\tau}{\delta}.$$

Let us check Assumption (K1). We have, using (18),

$$\|\mathrm{Jac}\, G(\theta^\star)^{-1} G(\theta^\star)\| \leq \frac{\|G(\theta^\star)\|}{\rho_{\min}(\mathrm{Jac}\, G(\theta^\star))} \leq \frac{\|p(\theta^\star)\|}{\delta} \leq \frac{\tau}{\delta}.$$

Hence (K1) holds since $\frac{\widetilde{R}}{2} = \frac{\tau}{\delta}$.

Let us check Assumption(K2). For all $\theta, \theta' \in B(\theta^\star, \widetilde{R})$, we have

$$\rho_{\max}\left(\operatorname{Jac} G(\theta^\star)^{-1}\left(\operatorname{Jac} G(\theta) - \operatorname{Jac} G(\theta')\right)\right) \leq \frac{1}{\rho_{\min}(\operatorname{Jac} G(\theta^\star))}\rho_{\max}\left(\operatorname{Jac} G(\theta) - \operatorname{Jac} G(\theta')\right)$$

$$\text{from (18), (19) and (20)} \quad \leq \frac{2M}{\delta}\|\theta - \theta'\|.$$

From (23), we have $\frac{\tau}{\delta} < \frac{\delta}{4M}$ and thus $\frac{2M}{\delta} \leq \frac{\delta}{2\tau} = \frac{1}{R}$. Hence (K2) holds.

Hence we can indeed apply Theorem 4 and we obtain that there is a unique $\theta_p \in B(\theta^\star, \frac{2\tau}{\delta})$ such that $G(\theta_p) = 0$, that is $F(\theta_p) = p(\theta_p)$ $\qquad\square$

*Proof of Theorem 1.*
Set $\theta^\star \in \operatorname{crit} L_1 \cap K$. We apply Proposition 6 with $F = \nabla L_1$, $p = -\nabla L_0$ and with $\theta^\star, M, \tau$ there given by the same notation here. We take $\delta$ there as $\delta/2$ here. Then indeed $F(\theta^\star) = 0$.

We have $\rho_{\min}(\nabla^2 L_1(\theta^\star)) \geq \delta$ from (1). Hence, using (5) and (6),

$$\rho_{\min}(\nabla^2(L_1 + L_0)(\theta^\star)) \geq \delta - \tau \geq \delta - \frac{\delta}{8} \geq \frac{\delta}{2}.$$

Hence (18) holds.

The conditions (19) to (22) hold by the assumptions (2) to (5). For (19) and (20), note that $B(\theta^\star, \frac{2\tau}{\delta/2}) \subset K$ from (7). The condition (23) holds from (6).

Hence all the assumptions of Proposition 6 are verified. We conclude that there is a unique $\theta_p \in B(\theta^\star, \frac{4\tau}{\delta})$ such that $F(\theta_p) = p(\theta_p)$, that is $\nabla(L_1 + L_0)(\theta_p) = 0$. From (7), $\theta_p \in B(\theta^\star, \frac{4\tau}{\delta}) \subset K$.

Assume now that there is a different number of strictly negative eigenvalues between $\nabla^2 L_1(\theta^\star)$ and $\nabla^2 L(\theta_p)$. Write $\lambda_1(Q) \leq \cdots \leq \lambda_m(Q)$ for the $m$ ordered eigenvalues of a symmetric $m \times m$ matrix $Q$. The eigenvalues of $\nabla^2 L_1(\theta^\star)$ are in $\mathbb{R}\backslash[-\delta, \delta]$ since we have observed that $\rho_{\min}(\nabla^2 L_1(\theta^\star)) \geq \delta$. Hence, if $\nabla^2 L_1(\theta^\star)$ and $\nabla^2(L_1 + L_0)(\theta_p)$ do not have the same number of strictly negative eigenvalues, there would exist $i \in \{1, \ldots, d\}$ such that $\left|\lambda_i(\nabla^2 L_1(\theta^\star)) - \lambda_i(\nabla^2(L_1 + L_0)(\theta_p))\right| \geq \delta$. However from Problem 4.3.P1 in [24], we have

$$\left|\lambda_i(\nabla^2 L_1(\theta^\star)) - \lambda_i(\nabla^2(L_1 + L_0)(\theta_p))\right| \leq \rho_{\max}\left(\nabla^2 L_1(\theta^\star) - \nabla^2(L_1 + L_0)(\theta_p)\right)$$

$$\leq \rho_{\max}\left(\nabla^2 L_1(\theta^\star) - \nabla^2 L_1(\theta_p)\right) + \rho_{\max}\left(\nabla^2 L_0(\theta_p)\right)$$

$$\text{from (2) and (5)} \quad \leq \frac{4\tau M}{\delta} + \tau$$

$$\text{from (6)} \quad \leq \frac{\delta}{8} + \frac{\delta}{8}$$

$$< \delta.$$

This is a contradiction and thus $\nabla^2 L_1(\theta^\star)$ and $\nabla^2(L_1 + L_0)(\theta_p)$ have the same number of strictly negative eigenvalues.

Consider now $\theta^\star \in \operatorname{crit}(L_1 + L_0) \cap K$. We will apply Proposition 6 with $F = \nabla L_1 + \nabla L_0$ and $p = \nabla L_0$. In Proposition 6 we will take for $\tau$ the same value as here. The quantity $M$ in Proposition 6 will be taken as $2M$ here. The quantity $\delta$ in Proposition 6 will be taken as $\delta/2$ here. Let us check the conditions of Proposition 6.

We have, from (4), and since $\nabla(L_1 + L_0)(\theta^\star) = 0$,

$$\|\nabla L_1(\theta^\star)\| \leq \tau \leq \frac{c}{2}.$$

Hence from (1), $\rho_{\min}(\nabla^2 L_1(\theta^\star)) \geq \delta$. Hence, from (5),

$$\rho_{\min}(\nabla^2(L_1 + L_0)(\theta^\star)) \geq \delta - \tau \geq \frac{\delta}{2} \tag{24}$$

because by assumption $\tau \leq \delta/8$. Hence (18) holds (with $\delta$ in (18) taken as $\delta/2$ here). Next, for all $\theta_1, \theta_2 \in B(\theta^\star, \frac{2\tau}{\delta/2}) \subset K$ (by (7)), from (2) and (3),

$$\rho_{\max}\left(\nabla^2(L_1 + L_0)(\theta_1) - \nabla^2(L_1 + L_0)(\theta_2)\right) \leq M\|\theta_1 - \theta_2\| + M\|\theta_1 - \theta_2\| = 2M\|\theta_1 - \theta_2\|. \tag{25}$$

Hence (19) holds (with $M$ in (19) taken as $2M$ here).

The conditions (20), (21) and (22) hold by assumption from (3), (4) and (5). For (20), note again that $B(\theta^\star, \frac{2\tau}{\delta/2}) \subset K$. Equation (23) in Proposition 6 holds from (6) in Theorem 1.

Hence the conclusion of Proposition 6 holds and there is $\theta_p \in B(\theta^\star, \frac{4\tau}{\delta})$ such that $\nabla(L_1+L_0)(\theta_p) = \nabla L_0(\theta_p)$, that is $\nabla L_1(\theta_p) = 0$. Also $\theta_p \in B(\theta^\star, \frac{4\tau}{\delta}) \subset K$.

Similarly as above, assume now that there is a different number of strictly negative eigenvalues between $\nabla^2(L_1+L_0)(\theta^\star)$ and $\nabla^2 L_1(\theta_p)$. Since we have established $\rho_{\min}(\nabla^2(L_1+L_0)(\theta^\star)) \geq \delta/2$ from (24), there would exist $i \in \{1,\ldots,d\}$ such that $\left|\lambda_i(\nabla^2(L_1+L_0)(\theta^\star)) - \lambda_i(\nabla^2 L_1(\theta_p))\right| \geq \delta/2$. However from Problem 4.3.P1 in [24], we have

$$
\begin{aligned}
\left|\lambda_i(\nabla^2(L_1+L_0)(\theta^\star)) - \lambda_i(\nabla^2 L_1(\theta_p))\right| &\leq \rho_{\max}\left(\nabla^2(L_1+L_0)(\theta^\star) - \nabla^2 L_1(\theta_p)\right) \\
&\leq \rho_{\max}(\nabla^2 L_0(\theta^\star)) + \rho_{\max}\left(\nabla^2 L_1(\theta^\star) - \nabla^2 L_1(\theta_p)\right) \\
\text{from (2) and (4)} \quad &\leq \tau + \frac{4\tau M}{\delta} \\
\text{from (6)} \quad &\leq \frac{\delta}{8} + \frac{\delta}{8} \\
&< \frac{\delta}{2}.
\end{aligned}
$$

This is a contradiction and thus $\nabla^2 L_1(\theta^\star)$ and $\nabla^2(L_1+L_0)(\theta_p)$ have the same number of strictly negative eigenvalues.

Hence we have established the theorem. $\qquad\square$

*Proof of Corollary 1.* From Theorem 1, for $\theta \in \text{crit } L_1 \cap K$, there is $\theta' \in \text{crit } L \cap K$ such that $\|\theta - \theta'\| \leq \frac{4\tau}{\delta}$. Conversely for $\theta \in \text{crit } L \cap K$, there is $\theta' \in \text{crit } L_1 \cap K$ such that $\|\theta - \theta'\| \leq \frac{4\tau}{\delta}$. Hence $\text{dist}_H(\text{crit } L_1 \cap K, \text{crit } L \cap K) \leq \frac{4\tau}{\delta}$.

Also from Theorem 1, for $\theta \in \text{argmin-loc } L_1 \cap K$, since $\theta \in \text{crit } L_1 \cap K$, there is $\theta' \in \text{crit } L \cap K$ such that $\|\theta - \theta'\| \leq \frac{4\tau}{\delta}$. Also $\nabla^2 L_1(\theta)$ has no strictly negative eigenvalues since $\theta \in \text{argmin-loc } L_1$. Hence from Theorem 1, $\nabla^2 L(\theta')$ has no strictly negative eigenvalues. As observed in (24) in the proof of Theorem 1, $\nabla^2 L(\theta')$ has no zero eigenvalues. Hence $\theta' \in \text{argmin-loc } L \cap K$. Similarly, for $\theta \in \text{argmin-loc } L \cap K$, we can show that there is $\theta' \in \text{argmin-loc } L_1 \cap K$ with $\|\theta - \theta'\| \leq \frac{4\tau}{\delta}$. Hence

$$
\text{dist}_H(\text{argmin-loc } L_1 \cap K, \text{argmin-loc } L \cap K) \leq \frac{4\tau}{\delta}.
$$

Finally, from (9), for each $\theta \in \text{argmin-loc } L_1 \cap K$, there is indeed $\theta' \in \text{argmin-loc } L \cap K$ with $\|\theta - \theta'\| \leq \frac{4\tau}{\delta}$. For each $\widetilde{\theta} \in B(\theta', \frac{6\tau}{\delta}) \subset K$, we have, using (24) and (25) from the proof of Theorem 1, and then (6),

$$
\lambda_{\min}\left(\nabla^2 L(\widetilde{\theta})\right) \geq \lambda_{\min}\left(\nabla^2 L(\theta')\right) - \rho_{\max}\left(\nabla^2 L(\widetilde{\theta}) - \nabla^2 L(\theta')\right) \geq \frac{\delta}{2} - \frac{12\tau M}{\delta} \geq \frac{\delta}{8}.
$$

This concludes the proof.

$\qquad\square$

**Corollary 2.** *In the context of Theorem 1, assuming further that* $\text{dist}_H(\text{crit } L \cap K, \text{bdry } K) \geq \frac{\delta}{M}$ *and* $\text{dist}_H(\text{crit } L_1 \cap K, \text{bdry } K) \geq \frac{\delta}{M}$, *we have the following additional conclusions.*

    *(i) For each pair of distinct elements* $\theta, \theta' \in \text{crit } L_1 \cap K$ *(resp.* $\text{crit } L \cap K$*), we have*

$$
\|\theta - \theta'\| \geq \frac{\delta}{32M}.
$$

    *(ii) The sets* $\text{crit } L_1 \cap K$ *and* $\text{crit } L \cap K$ *are finite.*

*Proof of Corollary 2.*

**Proof of the first conclusion.** To establish the first conclusion, consider $\theta^\star \in \text{crit } L_1 \cap K$. Let us apply Proposition 6 with $F = \nabla L_1$, $p$ taken as the zero function, $\delta$ there equal to $\delta$ here, $M$ there taken as $M$ here and $\tau$ there taken as a quantity that we write $\tau'$ and that is arbitrarily close to but strictly smaller than $\frac{\delta^2}{4M}$. With similar arguments as in the proof of Theorem 1, we can check that (18) holds, and that (19) holds. Trivially, (20), (21) and (22) hold. Finally (23) holds because

$$\frac{\tau'}{\delta} < \frac{\delta^2}{4M\delta} = \frac{\delta}{4M}.$$

Hence the conclusion of Proposition 6 is that for $\theta' \in \text{crit } L_1 \cap K$, $\theta' \neq \theta^\star$, we have

$$\|\theta^\star - \theta'\| \geq \frac{2\tau'}{\delta}.$$

Thus, letting $\tau'$ arbitrarily close to $\frac{\delta^2}{4M}$, we get

$$\|\theta^\star - \theta'\| \geq \frac{2}{\delta}\frac{\delta^2}{4M} = \frac{\delta}{2M} \geq \frac{\delta}{32M}.$$

Conversely, consider $\theta^\star \in \text{crit}\,(L_1 + L_0) \cap K$. Let us apply Proposition 6 with $F = \nabla(L_1 + L_0)$, $p$ taken as the zero function, $\delta$ there equal to $\delta/2$ here, $M$ there taken as $2M$ here and $\tau$ there taken as a quantity that we write $\tau'$ and that is arbitrarily close but strictly smaller to $\frac{\delta^2}{64M}$. With similar arguments as in the proof of Theorem 1, we can check that (18) holds, and that (19) holds. Trivially, (20), (21) and (22) hold. Finally (23) holds because

$$\frac{\tau'}{\frac{\delta}{2}} < \frac{2}{\delta}\frac{\delta^2}{64M} = \frac{\delta}{32M} \leq \frac{\frac{\delta}{2}}{4(2M)}.$$

Hence the conclusion of Proposition 6 is that for $\theta' \in \text{crit}\,(L_1 + L_0)$ with $\theta' \neq \theta^\star$, we have

$$\|\theta^\star - \theta'\| \geq 2\frac{\tau'}{\delta}.$$

Hence, letting $\tau'$ arbitrarily close to $\frac{\delta^2}{64M}$, we have

$$\|\theta^\star - \theta'\| \geq \frac{2}{\delta}\frac{\delta^2}{64M} = \frac{\delta}{32M}.$$

**Proof of the second conclusion.** Since $\text{crit } L_1 \cap K$ is bounded, and since to each $\theta \in \text{crit } L_1 \cap K$ we can associate a ball of fixed radius containing no other points of $\text{crit } L_1 \cap K$ (second conclusion), we deduce that $\text{crit } L_1 \cap K$ is a finite set. Similarly, $\text{crit}\,(L_1 + L_0) \cap K$ is a finite set. $\qquad\square$

## C.2 Proofs and extra results of Section 3.2

*Proof of Theorem 2.* Consider $\theta \in Z_{\text{maj-adv}} \cap K_{-\frac{2\tau}{\delta}}$. We have

$$\begin{aligned}
\langle \nabla L(\theta), \nabla L_1(\theta)\rangle =& \|\nabla L_1(\theta)\|^2 + \langle \nabla L_1(\theta), \nabla L_0(\theta)\rangle \\
\geq& \|\nabla L_1(\theta)\|^2 - \|\nabla L_1(\theta)\| \cdot \|\nabla L_0(\theta)\| \\
=& \|\nabla L_1(\theta)\| \left(\|\nabla L_1(\theta)\| - \|\nabla L_0(\theta)\|\right).
\end{aligned}$$

Note that, by (4), $\|\nabla L_0(\theta)\| \leq \tau$. Hence, since $\theta \in Z_{\text{maj-adv}}$, we have $\|\nabla L_1(\theta)\| \leq \tau$.

We then apply Theorem 4 with $\theta^\star$ there equal to $\theta$ here, with $G$ equal to $\nabla L_1$ and with $\widetilde{R}$ equal to $\frac{2\tau}{\delta}$. Note that $B(\theta, \widetilde{R}) \subset K$ by assumption. Since $\tau \leq c$ by (6), then from (1), we have $\rho_{\min}(\nabla^2 L_1(\theta)) \geq \delta$. Hence

$$\|(\nabla^2 L_1(\theta))^{-1}\nabla L_1(\theta)\| \leq \frac{\tau}{\delta}$$

and thus (K1) holds in Theorem 4. Also, for all $\theta', \theta'' \in B(\theta, \widetilde{R})$, we have from (2),

$$\rho_{\max}\left((\nabla^2 L_1(\theta))^{-1}\left(\nabla^2 L_1(\theta') - \nabla^2 L_1(\theta'')\right)\right) \leq \frac{M\|\theta' - \theta''\|}{\delta} \leq \frac{\|\theta' - \theta''\|}{\widetilde{R}},$$

because $\frac{M}{\delta} \le \frac{1}{R}$ since $\frac{2\tau}{\delta} \le \frac{\delta}{M}$ since $\tau \le \frac{\delta^2}{2M}$ from (6). Hence (K2) holds in Theorem 4.

Hence Theorem 4 implies that there exists $\widetilde{\theta}$ such that $\nabla L_1(\widetilde{\theta}) = 0$ and $\|\widetilde{\theta} - \theta\| \le \widetilde{R} = \frac{2\tau}{\delta}$. Hence we have
$$\theta \in \bigcup_{\widehat{\theta}_1 \in \text{crit } L_1 \cap K} B\left(\widehat{\theta}_1, \frac{2\tau}{\delta}\right)$$
which concludes the proof. □

*Proof of Lemma 1.* We have
$$\langle \nabla L(\theta), \nabla L_0(\theta) \rangle = \langle \nabla L(\theta), \nabla L(\theta) \rangle - \langle \nabla L(\theta), \nabla L_1(\theta) \rangle$$
$$\ge 0,$$
because $\theta \in Z_{\text{maj-adv}}$ means by definition that $\langle \nabla L(\theta), \nabla L_1(\theta) \rangle \le 0$. The rest is the classical Lyapunov computation. □

**Proposition 7.** *Consider that $L_0, L_1 : \mathbb{R}^d \to \mathbb{R}$ are twice continuously differentiable with the properties that*
$$\nabla(L_1(\theta)) = 0 \implies \rho_{\min}(\nabla^2 L_1(\theta)) > 0, \qquad \theta \in \mathbb{R}^d \tag{26}$$
*and, using $L = L_1 + L_0$,*
$$\nabla(L(\theta)) = 0 \implies \rho_{\min}(\nabla^2 L(\theta)) > 0, \qquad \theta \in \mathbb{R}^d. \tag{27}$$

*Then, recalling the symmetric difference notation*
$$\text{crit } L_1 \, \Delta \, \text{crit } L := (\text{crit } L_1 \cup \text{crit } L) \backslash (\text{crit } L_1 \cap \text{crit } L),$$
*we have*
$$\text{crit } L_1 \, \Delta \, \text{crit } L \subset \text{bdry } Z_{\text{maj-adv}}.$$

*Proof of Proposition 7.* Consider $\widehat{\theta}_1 \in \text{crit } L_1 \backslash \text{crit } L$. Then $\nabla L(\widehat{\theta}_1) \ne 0$. By continuity, there exists $\epsilon_0 > 0$ such that for $\|\theta - \widehat{\theta}_1\| \le \epsilon_0$,
$$\|\nabla L(\theta) - \nabla L(\widehat{\theta}_1)\| \le \frac{1}{2}\|\nabla L(\widehat{\theta}_1)\|.$$

Consider $0 < \epsilon < \epsilon_0$. From (26) and from the local inversion theorem, there are neighborhoods $U$ of $\widehat{\theta}_1$ and $V$ of $0 \in \mathbb{R}^d$ such that $U \subset B(\widehat{\theta}_1, \epsilon)$ and such that $\nabla L_1$ is bijective from $U$ to $V$.

Hence, there is $t_\epsilon > 0$ (small enough) and there is $\widetilde{\theta} \in U$ such that
$$\nabla L_1(\widetilde{\theta}) = t_\epsilon \nabla L(\widehat{\theta}_1) \in V. \tag{28}$$

Hence
$$\left\langle \nabla L_1(\widetilde{\theta}), \nabla L(\widetilde{\theta}) \right\rangle = t_\epsilon \left\langle \nabla L(\widehat{\theta}_1), \nabla L(\widetilde{\theta}) \right\rangle$$
$$= t_\epsilon \left\langle \nabla L(\widehat{\theta}_1), \nabla L(\widehat{\theta}_1)) \right\rangle + t_\epsilon \left\langle \nabla L(\widehat{\theta}_1), \nabla L(\widetilde{\theta}) - \nabla L(\widehat{\theta}_1) \right\rangle$$
$$\ge t_\epsilon \left\|\nabla L(\widehat{\theta}_1)\right\|^2 - t_\epsilon \left\|\nabla L(\widehat{\theta}_1)\right\| \cdot \left\|\nabla L(\widetilde{\theta}) - \nabla L(\widehat{\theta}_1)\right\|$$
$$\ge t_\epsilon \left\|\nabla L(\widehat{\theta}_1)\right\|^2 - t_\epsilon \left\|\nabla L(\widehat{\theta}_1)\right\| \cdot \frac{\left\|\nabla L(\widehat{\theta}_1)\right\|}{2}$$
$$> 0.$$

We can proceed similarly as from (28) but this time with $\widetilde{\theta}' \in U$, $t'_\epsilon < 0$ and
$$\nabla L_1(\widetilde{\theta}') = t'_\epsilon \nabla L(\widehat{\theta}_1) \in V.$$

This yields
$$\left\langle \nabla L_1(\widetilde{\theta}), \nabla L(\widetilde{\theta}) \right\rangle < 0.$$

Since this holds for any $\epsilon > 0$ there are two sequences $(\widetilde{\theta}_k)_k$ and $(\widetilde{\theta}'_k)_k$ that converge to $\widehat{\theta}_1$ with $\widetilde{\theta}_k \in Z^c_{\text{maj-adv}}$ and $\widetilde{\theta}'_k \in Z_{\text{maj-adv}}$. Hence $\widehat{\theta}_1 \in \text{bdry } Z_{\text{maj-adv}}$. Hence

$$\text{crit } L_1 \backslash \text{crit } L \subset \text{bdry } Z_{\text{maj-adv}}.$$

We can show symmetrically

$$\text{crit } L \backslash \text{crit } L_1 \subset \text{bdry } Z_{\text{maj-adv}}.$$

$\square$

## C.3 Proofs and extra results of Section 3.3

**Lemma 3** (Duration for proximity to a local minimizer). *Consider a function $L : \mathbb{R}^d \to \mathbb{R}$ that is twice continuously differentiable. Assume that there exists a trajectory $[0, \infty) \ni t \mapsto \theta(t)$ satisfying*

$$\frac{d}{dt}\theta(t) = -(\nabla L)(\theta(t)) \text{ with } \theta(0) = \theta_{\text{init}} \in \mathbb{R}^d.$$

*Consider a critical point $\widehat{\theta}$ of $L$ such that $\widehat{\theta} \neq \theta_{\text{init}}$. Consider a ball $\mathcal{B}$ containing $\widehat{\theta}$ and $\{\theta(t); t \geq 0\}$. Assume that, for some $M < \infty$ and for all $\theta \in \mathcal{B}$,*

$$\rho_{\max}\left(\nabla^2 L(\theta)\right) \leq M. \tag{29}$$

*Consider $\epsilon \in (0,1)$ and $t_\epsilon \in (0, \infty)$ satisfying*

$$\|\theta(t_\epsilon) - \widehat{\theta}\| \leq \epsilon \|\theta_{\text{init}} - \widehat{\theta}\|.$$

*Then we have*

$$t_\epsilon \geq \frac{1}{M} \log\left(\frac{1}{\epsilon}\right).$$

*Proof of Lemma 3.* Without loss of generality, we can consider that

$$t_\epsilon = \inf\left\{t \geq 0; \|\theta(t) - \widehat{\theta}\| \leq \epsilon\|\theta_{\text{init}} - \widehat{\theta}\|\right\} < \infty.$$

Consider the function $[0, t_\epsilon) \ni u \mapsto g(u) = \|\theta(t_\epsilon - u) - \widehat{\theta}\|$. Note that this function is strictly positive and differentiable on $[0, t_\epsilon]$ (since $\|\theta(t_\epsilon - u) - \widehat{\theta}\| \geq \epsilon\|\theta_{\text{init}} - \widehat{\theta}\| > 0$ for $u \in [0, t_\epsilon]$). The derivative at $u \in [0, t_\epsilon]$ satisfies

$$g'(u) = \left\langle \frac{d}{du}\theta(t_\epsilon - u), \frac{\theta(t_\epsilon - u) - \widehat{\theta}}{\|\theta(t_\epsilon - u) - \widehat{\theta}\|} \right\rangle$$

$$= \left\langle (\nabla L)(\theta(t_\epsilon - u)), \frac{\theta(t_\epsilon - u) - \widehat{\theta}}{\|\theta(t_\epsilon - u) - \widehat{\theta}\|} \right\rangle$$

$$\leq \|(\nabla L)(\theta(t_\epsilon - u))\|$$

$$= \|(\nabla L)(\theta(t_\epsilon - u)) - (\nabla L)(\widehat{\theta})\|$$

$$\text{Lemma 2:} \quad \leq M\|\theta(t_\epsilon - u) - \widehat{\theta}\|$$

$$= Mg(u).$$

Hence we can apply Grönwall's inequality, yielding

$$g(t_\epsilon) \leq g(0)e^{Mt_\epsilon} = \epsilon\|\theta_{\text{init}} - \widehat{\theta}\|e^{Mt_\epsilon}.$$

On the other hand $g(t_\epsilon) = \|\theta_{\text{init}} - \widehat{\theta}\|$ and thus

$$\epsilon\|\theta_{\text{init}} - \widehat{\theta}\|e^{Mt_\epsilon} \geq \|\theta_{\text{init}} - \widehat{\theta}\|.$$

This yields

$$t_\epsilon \geq \frac{1}{M}\log\left(\frac{1}{\epsilon}\right).$$

This concludes the proof. $\square$

*Proof of Proposition 1.* We apply Lemma 3 with $L$ in the lemma equal to $L$ here, with $M$ in the lemma equal to $M$ here and with $\epsilon$ in the lemma equal to

$$\frac{\|\widehat{\theta}_{\text{stereotype}} - \widehat{\theta}\|}{\|\theta_{\text{init}} - \widehat{\theta}\|}$$

here. Then we have

$$\|\widehat{\theta}_{\text{stereotype}} - \widehat{\theta}\| = \epsilon \|\theta_{\text{init}} - \widehat{\theta}\|$$

and so the lemma yields

$$t_{\text{stereotype}} \geq \frac{1}{M} \log\left(\frac{1}{\epsilon}\right)$$

which concludes the proof. □

*Proof of Proposition 2.* We apply Lemma 3 which directly concludes the proof. □

### C.4 Proofs and extra results of Appendix B

*Proof of Theorem 3.* Let us apply Proposition 6 to the linear model. We let, for $i = 0, 1$, $f_i(\theta) = \|Y^i - X^i \theta\|^2$. We can apply Proposition 6 to

$$F = \nabla f_1, \quad p = -\nabla f_0, \quad \theta^\star = \widehat{\theta}_1$$

and with constants $\delta, M, \tau$ to be specified later.

We have

$$\nabla f_i(\theta) = -2 X^{i\top} Y^i + 2 X^{i\top} X^i \theta$$

and

$$\nabla^2 f_i(\theta) = 2 X^{i\top} X^i.$$

Hence taking

$$\delta = 2 \rho_{\min}(X^{1\top} X^1)$$

we obtain that (18) holds in Proposition 6. Furthermore, the Hessian matrices of $f_0$ and $f_1$ are constant and thus we can take $M = 0$ in Proposition 6 while still having that (19) and (20) hold.

Next,

$$\begin{aligned}
\nabla f_0(\widehat{\theta}_1) &= -2 X^{0\top} Y^0 + 2 X^{0\top} X^0 \widehat{\theta}_1 \\
&= -2 X^{0\top} Y^0 + 2 X^{0\top} X^0 \widehat{\theta}_0 + 2 X^{0\top} X^0 (\widehat{\theta}_1 - \widehat{\theta}_0) \\
&= 2 X^{0\top} X^0 (\widehat{\theta}_1 - \widehat{\theta}_0).
\end{aligned}$$

Hence we take

$$\tau = 2 \rho_{\max}(X^{0\top} X^0)\left(1 + \|\widehat{\theta}_1 - \widehat{\theta}_0\|\right)$$

to ensure that $\rho_{\max}(\nabla^2 f_0(\widehat{\theta}_1)) \leq \tau$ and $\|\nabla f_0(\widehat{\theta}_1)\| \leq \tau$. Thus, (21) and (22) hold in Proposition 6. Hence, we can apply Proposition 6 that yields

$$\|\widehat{\theta} - \theta_1\| \leq \frac{2\tau}{\delta} = \frac{2\rho_{\max}(n_0 S_0)}{\rho_{\min}(n_1 S_1)}\left(1 + \|\widehat{\theta}_1 - \widehat{\theta}_0\|\right).$$

Note that the constraint (23) becomes vacuous since $M = 0$. □

The following lemma provides the expression of the (well-known) solutions of

$$\frac{d}{dt}\theta(t) = -\nabla L(\theta(t)), \quad \frac{d}{dt}\theta_i(t) = -\nabla L_i(\theta_i(t)), \ i = 1, 2, \quad \theta(0) = \theta_0(0) = \theta_1(0) = \theta_{\text{init}}.$$

Note that, with the uniqueness assumption, we have $\widehat{\theta} = (nS)^{-1} X^\top Y$ and $\widehat{\theta}_i = (n_i S_i)^{-1} X^{i\top} Y^i$.

**Lemma 4.** *We have, for $t \geq 0$,*

$$\theta(t) = \widehat{\theta} + e^{-tS} \left( \theta_{\text{init}} - \widehat{\theta} \right)$$

*and for $i = 0, 1$ and $t \geq 0$,*

$$\theta_i(t) = \widehat{\theta}_i + e^{-t(n_i/n)S_i} \left( \theta_{\text{init}} - \widehat{\theta}_i \right).$$

*Proof of Lemma 4.* We have

$$\begin{aligned}
\frac{d}{dt}\theta(t) &= -(\nabla L)(\theta(t)) \\
&= -\frac{1}{n}X^\top X\theta(t) + \frac{1}{n}X^\top Y \\
&= -\frac{1}{n}X^\top X\theta(t) + \frac{1}{n}X^\top X(X^\top X)^{-1}X^\top Y \\
&= -S\theta(t) + S\widehat{\theta}.
\end{aligned}$$

Hence,

$$\frac{d}{dt}\left( \theta(t) - \widehat{\theta} \right) = -S\left( \theta(t) - \widehat{\theta} \right)$$

and $\theta(0) - \widehat{\theta} = \theta_{\text{init}} - \widehat{\theta}$. Hence

$$\theta(t) = \widehat{\theta} + e^{-tS} \left( \theta_{\text{init}} - \widehat{\theta} \right).$$

We then provide a similar proof for $\theta_i(t)$. We have

$$\begin{aligned}
\frac{d}{dt}\theta_i(t) &= -(\nabla L_i)(\theta(t)) \\
&= -\frac{1}{n}X^{i\top} X^i\theta(t) + \frac{1}{n}X^{i\top}Y^i \\
&= -\frac{1}{n}X^{i\top} X^i\theta(t) + \frac{1}{n}X^{i\top} X^i(X^{i\top} X^i)^{-1}X^{i\top} Y^i \\
&= -\frac{n_i}{n}S_i\theta(t) + \frac{n_i}{n}S_i\widehat{\theta}_i.
\end{aligned}$$

Hence,

$$\frac{d}{dt}\left( \theta_i(t) - \widehat{\theta}_i \right) = -\frac{n_i}{n}S_i\left( \theta(t) - \widehat{\theta}_i \right)$$

and $\theta_i(0) - \widehat{\theta}_i = \theta_{\text{init}} - \widehat{\theta}_i$. Hence

$$\theta_i(t) = \widehat{\theta}_i + e^{-t\frac{n_i}{n}S_i} \left( \theta_{\text{init}} - \widehat{\theta}_i \right).$$

This concludes the proof. $\qquad\square$

*Proof of Proposition 3.* We have, using Lemma 4,

$$\begin{aligned}
\|\theta(t) - \theta_1(t)\| &\leq \left\| \left(I_d - e^{-tS}\right)\widehat{\theta} - \left(I_d - e^{-t\frac{n_1}{n}S_1}\right)\widehat{\theta}_1 \right\| + \left\| \left(e^{-tS} - e^{-t\frac{n_1}{n}S_1}\right)\theta_{\text{init}} \right\| \\
&= \left\| \left(I_d - e^{-tS}\right)\left(\widehat{\theta} - \widehat{\theta}_1\right) + \left(e^{-t\frac{n_1}{n}S_1} - e^{-tS}\right)\widehat{\theta}_1 \right\| + \left\| \left(e^{-tS} - e^{-t\frac{n_1}{n}S_1}\right)\theta_{\text{init}} \right\| \\
&\leq \rho_{\max}\left(I_d - e^{-tS}\right)\|\widehat{\theta} - \widehat{\theta}_1\| + \rho_{\max}\left(e^{-tS} - e^{-t\frac{n_1}{n}S_1}\right)\left(\|\widehat{\theta}_1\| + \|\theta_{\text{init}}\|\right).
\end{aligned}$$

Since $I_d - e^{-tS}$ has eigenvalues between 0 and 1 and from [34, Lemma 3.24], we obtain

$$\begin{aligned}
\|\theta(t) - \theta_1(t)\| &\leq \|\widehat{\theta} - \widehat{\theta}_1\| + t\rho_{\max}\left(S - \frac{n_1}{n}S_1\right)e^{-t\rho_{\min}\left(\frac{n_1}{n}S_1\right)}\left(\|\widehat{\theta}_1\| + \|\theta_{\text{init}}\|\right) \\
&= \|\widehat{\theta} - \widehat{\theta}_1\| + t\rho_{\max}\left(\frac{n_0}{n}S_0\right)e^{-t\rho_{\min}\left(\frac{n_1}{n}S_1\right)}\left(\|\widehat{\theta}_1\| + \|\theta_{\text{init}}\|\right).
\end{aligned}$$

The maximizer (over $t$) of $te^{-t\rho_{\min}\left(\frac{n_1}{n}S_1\right)}$ is $t_{\max} = 1/\rho_{\min}\left(\frac{n_1}{n}S_1\right)$ which yields

$$\sup_{t>0} \|\theta(t) - \theta_1(t)\| \leq \|\widehat{\theta} - \widehat{\theta}_1\| + \frac{\rho_{\max}(n_0 S_0)(\|\widehat{\theta}_1\| + \|\theta_{\text{init}}\|)}{e \cdot \rho_{\min}(n_1 S_1)}.$$

This concludes the proof. $\qquad\square$

*Proof of Proposition 4.* Without loss of generality, we can consider that $t_{\text{stereotype}} = 0$ and $\theta_{\text{init}} = \widehat{\theta}_1$. Because for $t \geq 0$, $\theta(t) \in Z_{\text{maj-adv}}$, the function $t \mapsto L_1(\theta(t))$ is non-decreasing. Assume that there exists $t < \infty$ such that $L_0(\theta(t)) = L_0(\widehat{\theta})$. Then $L(\theta(t)) \leq L(\widehat{\theta})$ and thus $\theta(t) = \widehat{\theta}$. From Lemma 4, this is a contradiction because $\theta_{\text{init}} \neq \widehat{\theta}$. Hence, because $L_0(\theta(0)) > L_0(\widehat{\theta})$, the function $t \mapsto L_0(\theta(t)) - L_0(\widehat{\theta})$ is strictly positive on $[0, \infty)$ by continuity.

Finally, for simplicity, write $t_\epsilon = t'_{\text{catchup},\epsilon}$. Assume that $t_\epsilon$ does not go to infinity as $\epsilon \to 0$. Then there is a subsequence $(\epsilon_\ell)_{\ell \in \mathbb{N}}$ going to zero and a constant $T < \infty$ such that $t_{\epsilon_\ell} \leq T$. By compacity, we can extract a further convergent subsequence $(t_{\epsilon_{\ell_k}})_{k \in \mathbb{N}}$ with $t_{\epsilon_{\ell_k}} \to t^\star \in [0, T]$. We have

$$\frac{L_0(\theta(t_{\epsilon_{\ell_k}})) - L_0(\widehat{\theta})}{L_0(\widehat{\theta}_1) - L_0(\widehat{\theta})} \leq \epsilon_{\ell_k} \xrightarrow[k \to \infty]{} 0$$

and thus by continuity $L_0(\theta(t^\star)) = L_0(\widehat{\theta})$. This is a contradiction, which concludes the proof. $\qquad\square$

*Proof of Proposition 5.* Let

$$\theta(t) = \widehat{\theta} + e^{-tS}\left(\widehat{\theta}_0 - \widehat{\theta}\right).$$

Then as in Lemma 4, we have

$$\begin{aligned}
\frac{d}{dt} L_0(\theta(t)) &= \left\langle (\nabla L_0)(\theta(t)), \frac{d}{dt}\theta(t) \right\rangle \\
&= \left\langle (\nabla L_0)(\theta(t)), -(\nabla L)(\theta(t)) \right\rangle \\
&= -\mathcal{S}(\theta(t)), \qquad\qquad (30)
\end{aligned}$$

defining

$$\mathcal{S}(\theta) = \left\langle \nabla L_0(\theta), \nabla L(\theta) \right\rangle.$$

Let

$$T = \frac{1}{\rho_{\min}(S)}.$$

Then

$$\|\theta(T) - \widehat{\theta}\| \leq e^{-T\rho_{\min}(S)}\|\widehat{\theta}_0 - \widehat{\theta}\| \leq \frac{\|\widehat{\theta}_0 - \widehat{\theta}\|}{2}.$$

Then, using Lemma 2, for any $\widetilde{\theta}$ in the segment between $\theta(T)$ and $\widehat{\theta}$,

$$\begin{aligned}
\|\nabla L_0(\widetilde{\theta})\| &= \|\frac{n_0}{n} S_0(\widetilde{\theta} - \widehat{\theta}_0)\| \\
\text{(convexity of Euclidean norm:)} \quad &\leq \frac{n_0}{n}\rho_{\max}(S_0)\left(\|\theta(T) - \widehat{\theta}_0\| + \|\widehat{\theta} - \widehat{\theta}_0\|\right) \\
&\leq \frac{n_0}{n}\rho_{\max}(S_0)\left(\|\theta(T) - \widehat{\theta}\| + 2\|\widehat{\theta} - \widehat{\theta}_0\|\right) \\
&\leq 3\frac{n_0}{n}\rho_{\max}(S_0)\|\widehat{\theta} - \widehat{\theta}_0\|.
\end{aligned}$$

Then, by convexity,

$$\begin{aligned}
L_0(\theta(T)) - L_0(\widehat{\theta}_0) &\geq \frac{n_0}{2n}\rho_{\min}(S_0)\|\theta(T) - \widehat{\theta}_0\|^2. \\
&\geq \frac{n_0}{8n}\rho_{\min}(S_0)\|\widehat{\theta} - \widehat{\theta}_0\|^2,
\end{aligned}$$

since $\|\theta(T) - \widehat{\theta}\| \leq \|\widehat{\theta} - \widehat{\theta}_0\|/2$. Also, using (30),

$$L_0(\theta(T)) - L_0(\widehat{\theta}_0) = \int_0^T \frac{dL_0(\theta(t))}{dt} dt = \int_0^T -S(\theta(t))dt \leq -T \min_{t \in [0,T]} S(\theta(t)).$$

Combining the two last displays,

$$\begin{aligned}
\min_{t \in [0,T]} \mathcal{S}(\theta(t)) \leq & \frac{L_0(\widehat{\theta}_0) - L_0(\theta(T))}{T} \\
\leq & -\frac{\frac{n_0}{n}\rho_{\min}(S_0)\|\widehat{\theta} - \widehat{\theta}_0\|^2}{8T} \\
= & -\frac{n_0}{8n}\rho_{\min}(S_0)\rho_{\min}(S)\|\widehat{\theta} - \widehat{\theta}_0\|^2.
\end{aligned}$$

Let $\overline{\theta} = \theta(\overline{t})$ with $\overline{t} \in \operatorname*{argmin}_{t \in [0,T]} S(\theta(t))$. For $\theta \in B(\overline{\theta}, R)$, we have

$$\begin{aligned}
\left|\mathcal{S}(\theta) - \mathcal{S}(\overline{\theta})\right| = & \left|\left\langle \nabla L(\theta), \nabla L_0(\theta)\right\rangle - \left\langle \nabla L(\overline{\theta}), \nabla L_0(\overline{\theta})\right\rangle\right| \\
= & \left|\left\langle \nabla L(\theta), \nabla L_0(\theta) - \nabla L_0(\overline{\theta})\right\rangle + \left\langle \nabla L(\theta) - \nabla L(\overline{\theta}), \nabla L_0(\overline{\theta})\right\rangle\right| \\
\text{(Lemma 2:)} \quad \leq & \|\nabla L(\theta)\|\frac{n_0}{n}\rho_{\max}(S_0)\|\theta - \overline{\theta}\| + \rho_{\max}(S)\|\theta - \overline{\theta}\|\|\nabla L_0(\overline{\theta})\| \\
\leq & R\frac{n_0}{n}\rho_{\max}(S_0)\|\nabla L(\theta)\| + R\rho_{\max}(S)\|\nabla L_0(\overline{\theta})\| \\
\text{(Lemma 2:)} \quad \leq & R\frac{n_0}{n}\rho_{\max}(S_0)\rho_{\max}(S)\|\theta - \widehat{\theta}\| + R\rho_{\max}(S)\frac{n_0}{n}\rho_{\max}(S_0)\|\overline{\theta} - \widehat{\theta}_0\| \\
\leq & R\frac{n_0}{n}\rho_{\max}(S_0)\rho_{\max}(S)\left(R + \|\overline{\theta} - \widehat{\theta}\| + \|\overline{\theta} - \widehat{\theta}_0\|\right).
\end{aligned}$$

We recall

$$\theta(t) = \widehat{\theta} + e^{-tS}\left(\widehat{\theta}_0 - \widehat{\theta}\right).$$

Hence, $\|\theta(t) - \widehat{\theta}\| \leq \|\widehat{\theta}_0 - \widehat{\theta}\|$ and $\|\theta(t) - \widehat{\theta}_0\| \leq 2\|\widehat{\theta}_0 - \widehat{\theta}\|$. Thus we have

$$\left|\mathcal{S}(\theta) - \mathcal{S}(\overline{\theta})\right| \leq R\frac{n_0}{n}\rho_{\max}(S_0)\rho_{\max}(S)\left(R + 3\|\widehat{\theta}_0 - \widehat{\theta}\|\right).$$

Hence, let us take $R$ as in (16), with in particular $R \leq \|\widehat{\theta}_0 - \widehat{\theta}\|$. Then to satisfy $\langle \nabla L(\theta), \nabla L_0(\theta)\rangle \leq 0$ for all $\theta \in B(\overline{\theta}, R)$, it is sufficient that

$$R\frac{n_0}{n}\rho_{\max}(S_0)\rho_{\max}(S)\left(R + 3\|\widehat{\theta}_0 - \widehat{\theta}\|\right) < \frac{1}{8}\frac{n_0}{n}\rho_{\min}(S_0)\rho_{\min}(S)\|\widehat{\theta} - \widehat{\theta}_0\|^2.$$

For this it is sufficient that

$$32R\frac{n_0}{n}\rho_{\max}(S_0)\rho_{\max}(S)\|\widehat{\theta}_0 - \widehat{\theta}\| < \frac{n_0}{n}\rho_{\min}(S_0)\rho_{\min}(S)\|\widehat{\theta} - \widehat{\theta}_0\|^2$$

which is implied by

$$R < \frac{\rho_{\min}(S_0)\rho_{\min}(S)}{32\rho_{\max}(S_0)\rho_{\max}(S)}\|\widehat{\theta} - \widehat{\theta}_0\|.$$

This concludes the proof. $\qquad\square$

# D   Additional experiments on CIFAR-10

In this section, in Figures 6 to 8, we provide complementary results for additional architectures on the Imbalanced CIFAR-10 benchmark. We report detailed training and test dynamics (loss and accuracy) across different subgroup imbalance levels ($\zeta \in \{1\%, 10\%, 30\%\}$) and threshold $\kappa = 90\%$. These figures illustrate that the qualitative behavior observed in the main paper is consistent across models of varying depth and capacity.

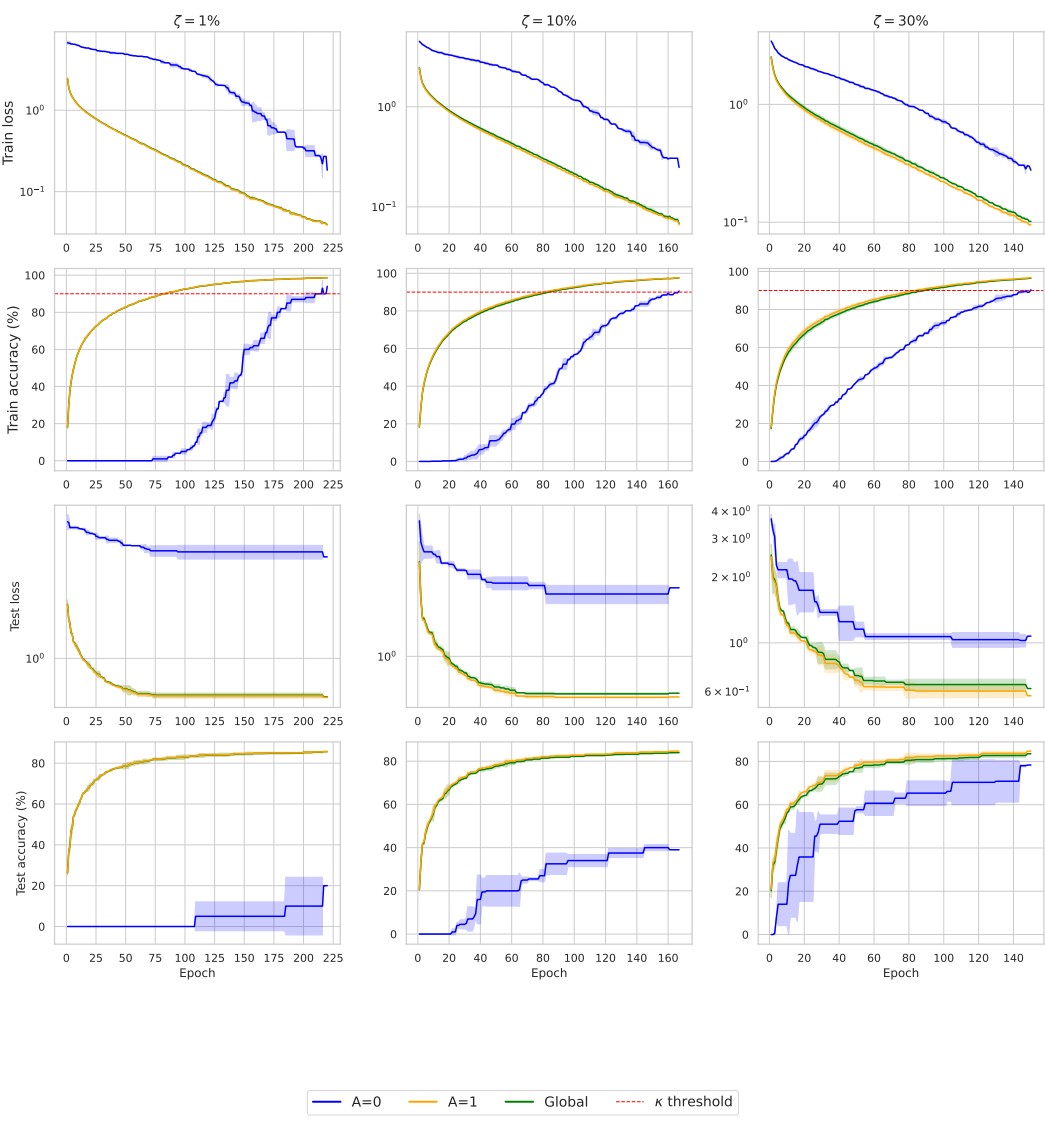

Figure 6: Training and test loss/accuracy dynamics for ResNet50 on CIFAR-10 across imbalance scenarios ($\zeta \in \{1\%, 10\%, 30\%\}$) with threshold $\kappa = 90\%$. Minority accuracy lags behind global and majority early, then catches up on both train and test. Compared to VGG-19 the delay to reach $\kappa$ is shorter, hence a lower catch-up overcost. Capacity mitigates, but does not remove, the extra training required for minority awareness.

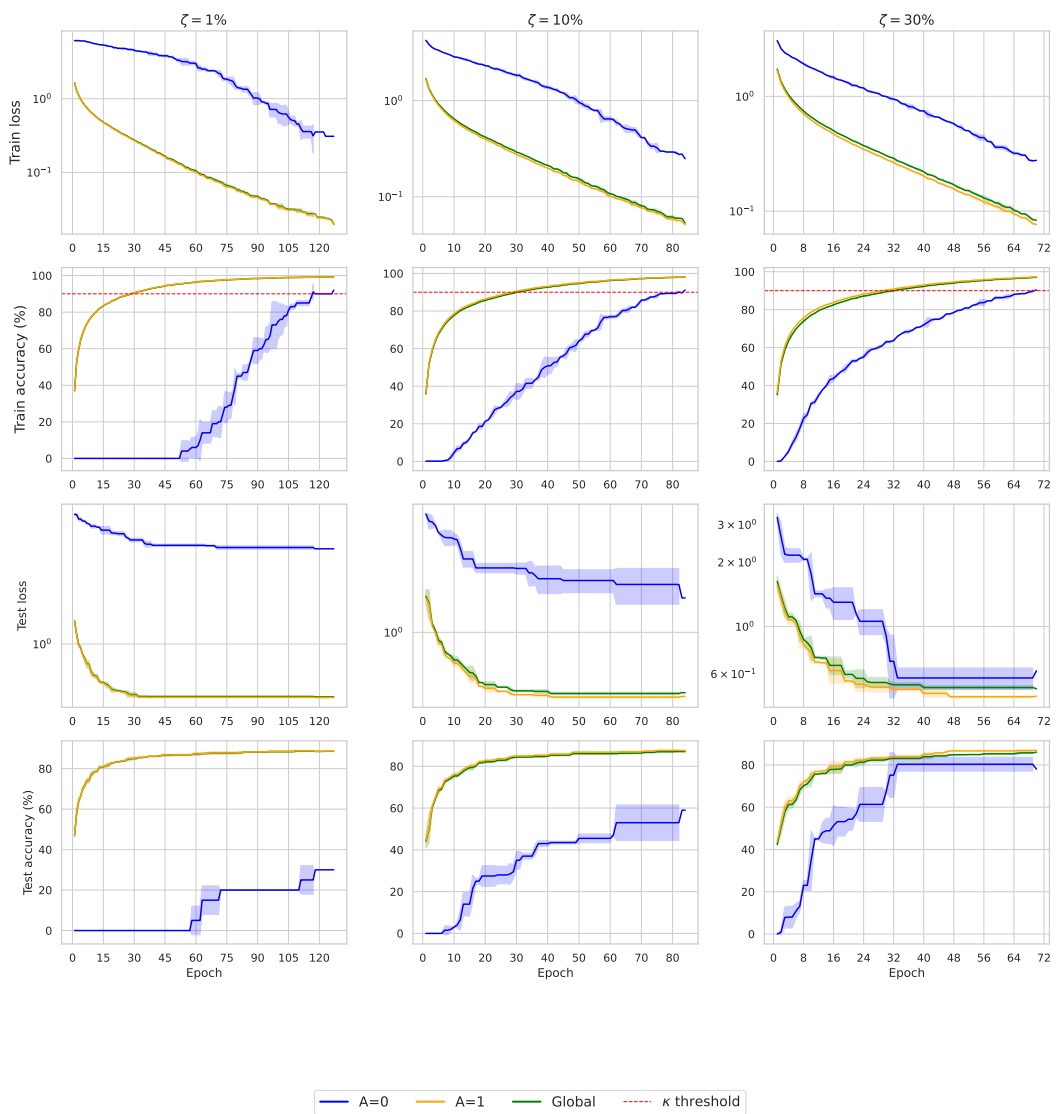

Figure 7: Training and test loss/accuracy dynamics for VGG19 on CIFAR-10 across imbalance scenarios ($\zeta \in \{1\%, 10\%, 30\%\}$) with threshold $\kappa = 90\%$. VGG19 is much bigger than ResNet-50 and does faster minority learning. However one sees a higher catch-up overcost (e.g. 280% VGG19 vs 157% ResNet50 for the 1% imbalance scenario) underscoring the role of architecture in the extra training needed to achieve minority awareness.

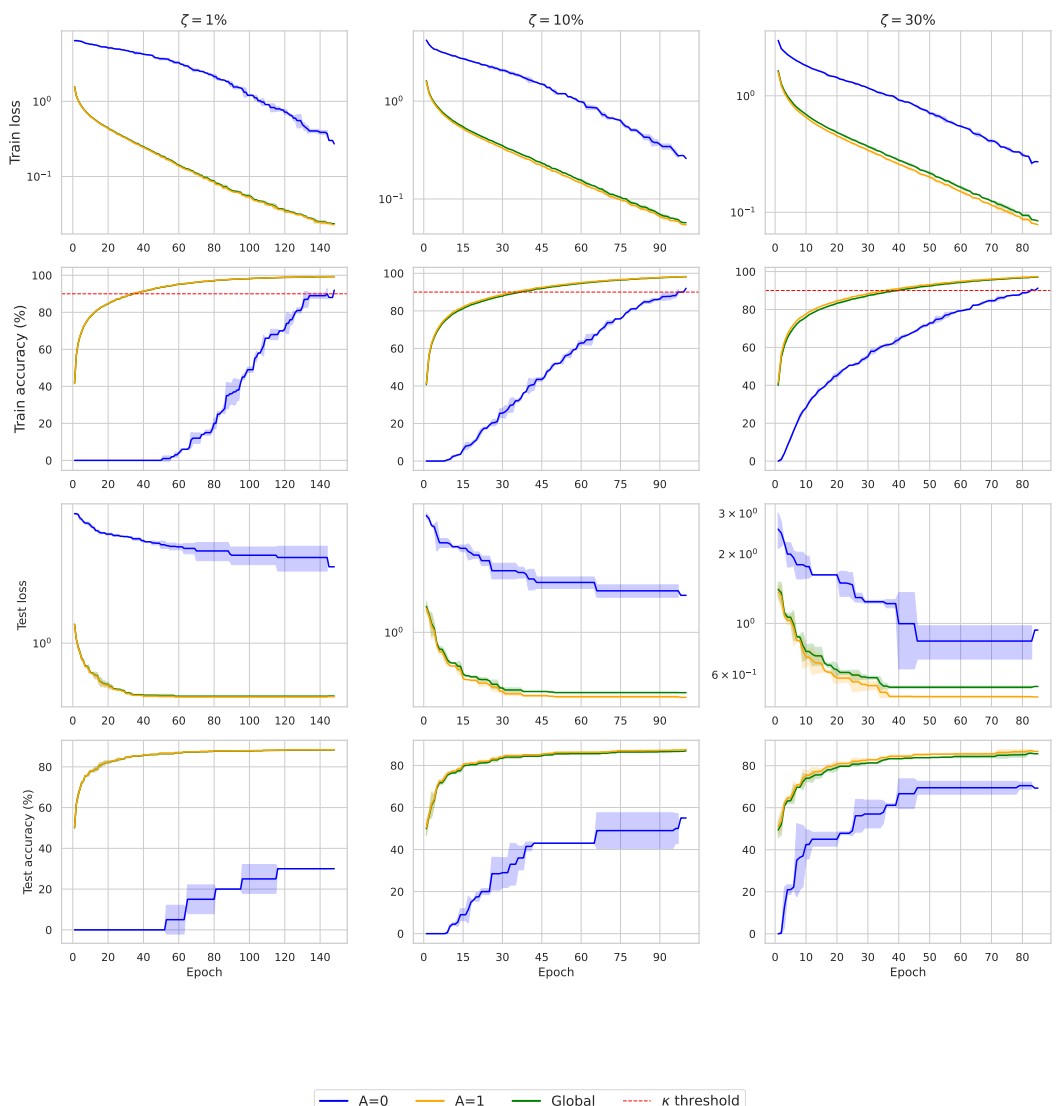

Figure 8: Training and test loss/accuracy dynamics for VGG11 on CIFAR-10 across imbalance scenarios ($\zeta \in \{1\%, 10\%, 30\%\}$) with threshold $\kappa = 90\%$. Training and test dynamics display a clear minority ($A = 0$) delay to the $\kappa$ threshold, longer than with ResNet-50 and comparable or slightly worse than VGG-19. Consistently with Table 2, the catch-up overcost at $\kappa = 90\%$ is high (e.g., $291\%$, $171\%$, $114\%$ for $\zeta = 1\%, 10\%, 30\%$), underscoring the role of architecture in the additional training required to attain minority awareness (cf. Figures 6 and 7).

### D.1 Impact of the optimizer: AdamW

While the main experiments in the paper use standard SGD without momentum, we also investigated the impact of an adaptive optimizer. In particular, we repeated the same Imbalanced CIFAR-10 protocol using AdamW with learning rate $\eta = 1 \times 10^{-3}$, weight decay $1 \times 10^{-2}$, and a cosine annealing scheduler. The loss function was standard cross-entropy.

Minority accuracy ($A = 0$) shows the same early delay observed with SGD: global and majority ($A = 1$) reach $\kappa$ first, and the minority catches up later on train and test. AdamW often reaches the global threshold sooner, yet the *catch-up overcost* remains of comparable magnitude —large under strong imbalance (about $400\%$ at $\zeta = 1\%$) and decreasing as $\zeta$ grows. As reported in Table 3, the overcost remains substantial across imbalance levels; changing the optimizer does not eliminate bias amplification, while its magnitude can vary with model capacity (see, e.g., [29]).

Table 3: Catch-up overcost (in $\%$) with AdamW on imbalanced CIFAR-10. Reported values are means over 3 independent runs across imbalance levels $\zeta \in \{1\%, 10\%, 30\%\}$.

| Model | Parameters | 1% | 10% | 30% |
|---|---|---|---|---|
| MobileNetV2 [38] | 543K | 326 | 310 | 214 |
| VGG11 [41] | 9M | 401 | 244 | 169 |
| ResNet18 [20] | 11M | 465 | 342 | 209 |
| ResNet50 | 25M | 369 | 327 | 219 |
| ResNet101 | 42M | 220 | 192 | 172 |

## E Additional experiments on Adult

To complement our deep learning results, we also ran an XGBoost logistic regression on Adult. The model was trained incrementally by adding batches of 10 trees at each iteration (using the `xgb_model` argument to continue training from the previous booster). Each step records cumulative training time and subgroup accuracies for $A = 0$ (minority) and $A = 1$ (majority). The optimizer and objective are handled internally by XGBoost (`eval_metric="logloss"`).

Despite the very different model class, we observe the same qualitative behavior: a pronounced minority delay to the $\kappa$ threshold, followed by a late catch-up visible on train and test. Changing the learning rate, tree depth, or regularization alters the number of boosting rounds needed to reach the threshold, but the *relative* catch-up overcost remains large under strong imbalance (about $400\%$ at $\zeta = 1\%$) and declines as $\zeta$ grows. This shows the phenomenon is model-class robust, extending beyond deep networks.

Table 4: Incremental training with XGBoost logistic regression on Adult. We report global accuracy and subgroup accuracies for $A = 0$ (minority) and $A = 1$ (majority) on both train and test sets as the number of trees increases.

| Training time (s) | Train accuracy | | | Test accuracy | | |
|---|---|---|---|---|---|---|
| | Global | $A = 0$ | $A = 1$ | Global | $A = 0$ | $A = 1$ |
| 0.47 | 0.8772 | 0.5530 | 0.8893 | 0.8679 | 0.5263 | 0.8807 |
| 7.70 | 0.8989 | 0.6734 | 0.9074 | 0.8679 | 0.5996 | 0.8781 |
| 16.36 | 0.9098 | 0.7235 | 0.9168 | 0.8668 | 0.5940 | 0.8771 |
| 26.45 | 0.9190 | 0.7623 | 0.9249 | 0.8649 | 0.5921 | 0.8752 |
| 38.03 | 0.9261 | 0.7987 | 0.9308 | 0.8624 | 0.5977 | 0.8724 |
| 51.11 | 0.9312 | 0.8205 | 0.9353 | 0.8606 | 0.5996 | 0.8705 |
| 65.64 | 0.9359 | 0.8464 | 0.9393 | 0.8589 | 0.5959 | 0.8688 |
| 82.94 | 0.9385 | 0.8529 | 0.9417 | 0.8579 | 0.5959 | 0.8678 |

# F   Experimental details

This section provides full details to ensure reproducibility of our experiments. We describe hardware specifications, training hyperparameters, implementation details, and evaluation protocol for each dataset and model used.

## F.1   Datasets

We conduct experiments on a mix of image and tabular datasets with varying levels of class imbalance. Below, we describe the construction and preprocessing steps for each dataset used in our study.

**CIFAR-10.**   We use the standard CIFAR-10 dataset, consisting of 60,000 color images (32×32 pixels) in 10 classes, with 50,000 training and 10,000 test samples. To induce group imbalance, we define a binary sensitive attribute $A \in 0, 1$, following the approach detailed in Section 4.

**CIFAR-2.**   We consider a binary classification task derived from CIFAR-10 by selecting the two vehicle-related classes "automobile" and "truck". We refer to this subset as CIFAR-2. To simulate a highly imbalanced scenario, we drastically reduce the number of "automobile" (car) samples to a small fraction of their original count (e.g., retaining only 3%), while keeping all "truck" examples. This creates a pronounced majority-minority setting, suitable for studying bias amplification under imbalance.

**EuroSAT.**   EuroSAT is a land use and land cover classification dataset based on Sentinel-2 satellite images. We use the RGB version comprising 27,000 labeled images across 10 classes. For our binary classification task, we select two visually distinct classes: *Highway* and *River*. The input images are resized to 64×64 pixels and normalized. We define a binary sensitive attribute $A$ by thresholding the average blue-channel intensity to distinguish between "bluish" and "non-bluish" images, following the approach of [36].

**Adult.**   The Adult dataset is a standard benchmark for fairness and tabular learning. It contains approximately 48,000 examples with demographic and income information. We treat the binary income variable as the label and use "gender" (male vs. female) as the sensitive attribute $A$.

## F.2   Hardware and runtime

Experiments were conducted on a computing cluster equipped with NVIDIA A100 40GB GPUs. Each experiment ran on a single GPU unless otherwise specified. Average runtime per training run is reported in Table 5.

Table 5: Average training time per run across datasets and models.

| Dataset | Model | Runtime (h) | GPU |
|---------|-------|-------------|-----|
| CIFAR-10 | ResNet-18 | 0.26 | A100 |
| EuroSAT | ResNet-18 | 0.1 | A100 |
| Adult | TabNet | 0.16 | A100 |

## F.3   Optimization and training

We use SGD with a constant learning rate for image models and tabular data. In order to match our theoretical setting, no weight decay or learning rate decay schedule was applied. Models were trained from scratch without pretaining. Refer to Table 6 for more details.

## F.4   Evaluation and reporting

All results are averaged over 3 random seeds. We report mean and standard deviation of accuracy and loss metrics across groups. Class imbalance ratios $\zeta$ are detailed in the main text (Section 4).

Table 6: Optimization hyperparameters for each task.

| Dataset | Model | Optimizer | Learning rate |
|---------|-------|-----------|---------------|
| CIFAR-10 | ResNet-18 | SGD | $1 \times 10^{-2}$ |
| CIFAR-10 | VGG19 | SGD | $1 \times 10^{-2}$ |
| EuroSAT | ResNet-18 | SGD | $1 \times 10^{-4}$ |
| Adult | TabNet | SGD | $2 \times 10^{-2}$ |

## F.5 Reproducibility

All code and configuration files (including seed control, training logs, and plotting scripts) are available at `https://github.com/ryanboustany/bias_amplification`. We follow best practices for reproducible research and ensure all experimental figures can be regenerated with a single command.

