# OpenReview forum: "When majority rules, minority loses: bias amplification of gradient descent"
_NeurIPS.cc/2025/Conference — NeurIPS 2025 poster_

### Official Review · Reviewer_4HDc · 2025-06-30

**Clarity:** 2
**Significance:** 3
**Originality:** 2
**Rating:** 4
**Confidence:** 3

**Summary:**

The paper develops a framework for majority-minority learning tasks, going beyond empirical evidence of bias amplification to provide theoretical foundations. It shows how standard training can favor majority groups and produce stereotypical predictors that neglect minority-specific features.

**Questions:**

- In the experiments on the imbalanced CIFAR-10 dataset, the paper states that one class (A=0) is subsampled while others (A=1) remain unchanged. Which specific class is chosen? Does the choice of class influence the results?
- Does Theorem 2 hold only under the linear regression model assumption? Why was linear regression selected as the case study?
- Although the paper states that “assessing fairness on the test set would require extensive tuning and is left for future work,” and evaluates fairness only using training accuracy, are there any preliminary insights into the gap between training-set fairness and test-set fairness?

**Ethical Concerns:**

["NO or VERY MINOR ethics concerns only"]

**Final Justification:**

The response addressed my concerns and helped me better understand the work. After reading the reply and the comments from other reviewers, I lean towards acceptance.

**Limitations:**

yes

**Quality:**

2

**Strengths And Weaknesses:**

**Strengths**

- The paper conducts various experiments across image and tabular datasets to validate the effectiveness of the proposed method and introduces the "fairness overcost ratio" to quantify the additional computational cost required for fair training.
- Based on both theory and experiments, the paper offers practical advice: extend training duration, use larger models, and carefully monitor minority-group performance to avoid stereotype gaps.

**Weaknesses**

- The paper lacks a discussion of the related works. Please add a section on related works in the main paper.
- The theoretical part relies on relatively restrictive assumptions, such as in Theorem 1, which assumes, which assumes $\\tau < \\min\\left\\{ \\frac{c}{2}, \\frac{\\delta}{8}, \\frac{\\delta^2}{32M} \\right\\}$. Do the real-world models, like the models tested in the experiments, meet these assumptions?
- The theoretical results are derived primarily for convex/quadratic losses, while the experiments are conducted using non-convex deep networks. Is there a theoretical gap between these settings?
- The presentation could be improved.
    - For example, Figure 1 could be better illustrated with a more detailed caption or elsewhere in the paper, better pointing out what Figure 1 wants to show.
    - Section 2.4 (linear regression) appears abruptly after the general theoretical framework without a clear rationale. The authors don't justify why linear regression was chosen as the case study or explain what insights it provides beyond the general theory. This also creates poor logical flow: general theory → sudden linear case → back to general gradient dynamics.

---

> ### Author Rebuttal · Authors · 2025-07-30
>
> **General comment.**
>
>
> We greatly thank the referees for their useful comments, we believe that they will considerably improve the quality of the paper. We would like to make preliminary statements that will also impact the presentation of our revision.
>
> **A)** We will certainly include all the missing references —we apologize for the oversight, and we acknowledge their importance. A dedicated section on previous work will be added.
>
> **B)** Our contribution is principally theoretical and aims to explain the majority/minority scenario in the nonconvex setting. The empirical part is merely intended to illustrate our theorems, which is why it focuses solely on raw SGD. Our approach via the Kantorovich theorem is both non-trivial and notably robust to changes in assumptions, as it applies to nonlinear equations and admits well-known generalizations to Banach spaces and Riemannian manifolds. While we have opted for a reader-friendly presentation by placing most of the mathematical intricacies in the Appendix, these should not be overlooked, as they form the core of our contribution.
>
> **C)** Through our theoretical contributions, we also provide a framework for analyzing fair training by introducing several notions: the stereotypical gap, stereotypical predictors, fairness overcost, and a geometrical framework for understanding "stereotyped training," which can readily be adapted to other methods.
>
>
> We now proceed to answering point by point.
>
> **Reviewer 4HDc:** "*The paper lacks a discussion of the related works. Please add a section on related works in the main paper.*"
>
> **Authors:** We definitely agree, a section on related work will be added. We refer to our responses to Reviewer 48E9.
>
> **Reviewer 4HDc:** "*The theoretical part relies on relatively restrictive assumptions ... meet these assumptions?*"
>
> **Authors:** It seems to us that our assumptions are in line with common standards in Deep Learning, which often offer only qualitative insights into key quantitative aspects. In contrast, our assumptions are rather concrete: they amount to say that variability (embodied in second-order dominance) and population ratios must be unbalanced. This is further illustrated in the convex case (Theorem 2), which is of course more accessible. On this occasion, we should also mention that applying Theorem 1 to the convex case yields estimates very similar to those in Theorem 2—except for some boundedness constraints, which can be easily lifted by leveraging convexity.
>
> Note as well that, as explained to the other referees, strong imbalance is likely to provide the assumptions required by Theorem 1.
>
> We will add comments on our main assumptions and concrete unbalanced examples in which our setting makes sense concretely.
>
>
>
> **Reviewer 4HDc:** "*The theoretical results are derived primarily for convex/quadratic losses ... between these settings?*"
>
> **Authors:** We respectfully disagree with this statement, though we will revise the order of presentation to avoid such confusion. Our theoretical results are stated for general nonconvex losses—this includes Theorem 1, Corollary 1, Theorem 3, Lemma 1, and Propositions 2 and 3 (which require minor adjustments as noted in Remark 2 and the Appendix).
>
> The only results that are specific to convex or quadratic losses are Theorem 2—an “augmented” and specialized instance of Theorem 1—and Proposition 1, which are quadratic by construction. As clarified in Remark 2, Propositions 2 and 3 rely on Lemma 4, which remains valid in the nonconvex setting; they therefore hold under the usual localization arguments.
>
> In summary, we do not see a theoretical gap between the nonconvex and convex-quadratic settings. We have emphasized the convex-quadratic case simply to provide a more concrete and interpretable illustration, especially given the familiarity of many readers with that framework.
>
> **Reviewer 4HDc:** "*The presentation could be improved. For example, Figure 1 ... what Figure 1 wants to show.*"
>
> **Authors:** We agree with the comment on Figure 1. As also replied to Reviewer DEPM, we will explain that within the majority training zone on Figure 1, learning is almost blind to $L_0$, making little distinction between training with $L_1$ and $L$. The learning of $L_0$ is mainly activated outside or near the boundary of this zone. Here is part of the updated caption that we propose: The majority training zone, where only $L_1$ is effectively learned, covers almost the entire space. Other captions will be clarified.
>
> **Reviewer 4HDc:**  "*The presentation could be improved. Section 2.4 (linear regression) appears ... sudden linear case → back to general gradient dynamics.*"
>
> **Authors:** As stated above, our motivation for including linear regression is theoretical (it is the “linearized model” in the Morse case), it is an illustration of our theorem (improved through convexity), and it is also simpler and easily interpretable. But this comment, along with your earlier remark on the distinction between the nonconvex and convex-quadratic settings, prompts us to give less prominence to linear regression. We will isolate the results specific to linear regression in a dedicated section.
>
> **Reviewer 4HDc:** "*Question: In the experiments on the imbalanced CIFAR-10 dataset ... Does the choice of class influence the results?*"
>
> **Authors:** Thank you for your question. In our experiments on imbalanced CIFAR-10, we subsampled the class “cat” as the underrepresented group (A = 0), while keeping the other nine classes unchanged (A = 1). We chose “cat” arbitrarily to ensure reproducibility across runs, but we emphasize that the qualitative conclusions of our paper do not depend on this specific choice, as confirmed by our preliminary tests. We will clarify this point in the revised version.
>
>
> **Reviewer 4HDc:** "*Question: Does Theorem 2 hold only under the linear regression model assumption? Why was linear regression selected as the case study?*"
>
> **Authors:** Theorem 2 is indeed specialized to linear regression, but it is an instance of Theorem 1, which holds in the general nonconvex setting. The benefit of Theorem 2 is that it requires the definition of fewer quantities than Theorem 1 and offers a more accessible interpretation for a broader audience.
>
> Beyond the familiarity of linear regression to most readers, this model has a deeper theoretical significance: by Morse’s lemma and linearization theorems, the local geometry of Morse functions (our nonconvex assumption) is essentially quadratic—hence, locally equivalent to the linear regression case. The theoretical rationale is thus one of *local linearization*, which naturally leads to convex models, and in particular to linear regression.
>
>
> **Reviewer 4HDc:** "*Question: Although the paper states that ... between training-set fairness and test-set fairness?*"
>
> **Authors:** While our main results focus on the training set, we did run preliminary tests on the test set. The trends are similar overall, but test set disparities tend to be slightly larger, likely due to generalization gaps, especially for the minority group. A full analysis would require careful tuning, which we leave for future work.

---

> > ### Comment · Reviewer_4HDc · 2025-08-07
> >
> > Thank you for the detailed response, which addressed my concerns and helped me better understand the work. After reading your reply and the comments from other reviewers, I am willing to increase my score. Please ensure the final version is revised accordingly, especially for the missing related work.

---

> > > ### Author Response · Authors · 2025-08-08
> > >
> > > Thank you for your constructive feedback and for considering an increased score. We will update the final version to address the missing related work and the other points you raised.

---

> ### Author Response · Authors · 2025-08-06
>
> Dear reviewer,
>
> According to a recent message we received from the Program Chairs
> "Where reviewers have not responded yet to your rebuttal, you may initiate discussions with reviewers by yourself and ask the AC to help initiate such discussions",
> we would like to indicate our availability and willingness to discuss any point of our rebuttal that would benefit from clarifications.

---

### Official Review · Reviewer_48E9 · 2025-07-01

**Clarity:** 3
**Significance:** 2
**Originality:** 3
**Rating:** 4
**Confidence:** 2

**Summary:**

This paper investigates theoretical causes of *bias amplification*, specifically in settings where class imbalance leads models to favor majority groups over minority ones. The authors formalize a majority–minority learning scenario in which the empirical loss decomposes into majority loss $L_1$ and minority loss $L_0$, with $L_0 \ll L_1$. They prove a perturbation theorem: every critical point of the full loss $L = L_0 + L_1$ is close to a critical point of $L_1$ alone. These paired solutions are termed “representative” (trained on all data) and “stereotypical” (majority-only), and their distance, the “stereotype gap”, is bounded by a quantity that depends on the imbalance in population size and variance. This implies that most of parameter space is dominated by stereotypical solutions. The authors also derive a lower bound on the time required to escape these solutions during training. Experiments on tabular and vision tasks show that typical training procedures converge to stereotypical predictors, and that long training or dataset rebalancing is needed to recover features specific to the minority group.

**Questions:**

* Please expand the related work on the theory of bias amplification, including how this work connects to and extends prior theoretical and empirical studies.
* Please include a brief description of the main assumptions (e.g., in Theorem 1) and discuss when they are expected to hold in practice.

**Ethical Concerns:**

["NO or VERY MINOR ethics concerns only"]

**Final Justification:**

I believe the proposed theoretical framework offers a valuable perspective for understanding bias amplification. While the rebuttal provided a rough characterization of the main assumptions and included a commitment to incorporate the suggested related work, a clearer overview of all core assumptions is still lacking, and it remains unclear how the contribution fits within the broader context of existing literature. Overall, my recommendation is a borderline accept.

**Limitations:**

The authors adequately addressed the limitations.

**Paper Formatting Concerns:**

No paper formatting concerns.

**Quality:**

3

**Strengths And Weaknesses:**

**Strengths**

* The paper is generally well-written and clearly structured.
* It tackles an important and under-explored theoretical question: why models trained on imbalanced data tend to amplify bias against minority groups.
* The theoretical results are not purely abstract; they have practical implications, for instance, highlighting the potential risks of early stopping when validation metrics are dominated by majority-class performance.
* The experimental evaluation spans a diverse range of settings, including multiple model architectures, various imbalance ratios, and both image and tabular datasets.

**Weaknesses**

* Although the paper claims to be among the few theoretical treatments of bias amplification (lines 28–30), it overlooks several relevant prior works that directly address this issue from a theoretical standpoint \[1,2,3].
* Key assumptions underlying the theoretical results (e.g., those in Theorem 1) are not discussed in terms of their realism or the conditions under which they are likely to hold in practice.
* While the main theoretical insight is sound, the underlying intuition is fairly well-established in the field, so the paper may not significantly advance the state of the art.

References:

[1] Francazi, E., Baity-Jesi, M. and Lucchi, A., 2023, July. A theoretical analysis of the learning dynamics under class imbalance. In International Conference on Machine Learning (pp. 10285-10322). PMLR.

[2] Subramonian, A., Bell, S.J., Sagun, L. and Dohmatob, E., 2025. An Effective Theory of Bias Amplification. In International Conference on Learning Representations.

[3] Mannelli, S.S., Gerace, F., Rostamzadeh, N. and Saglietti, L., Bias-inducing geometries: exactly solvable data model with fairness implications. In ICML 2024 Workshop on Geometry-grounded Representation Learning and Generative Modeling.

---

> ### Author Rebuttal · Authors · 2025-07-30
>
> **General comment.**
>
>
> We greatly thank the referees for their useful comments, we believe that they will considerably improve the quality of the paper. We would like to make preliminary statements that will also impact the presentation of our revision.
>
> **A)** We will certainly include all the missing references —we apologize for the oversight, and we acknowledge their importance. A dedicated section on previous work will be added.
>
> **B)** Our contribution is principally theoretical and aims to explain the majority/minority scenario in the nonconvex setting. The empirical part is merely intended to illustrate our theorems, which is why it focuses solely on raw SGD. Our approach via the Kantorovich theorem is both non-trivial and notably robust to changes in assumptions, as it applies to nonlinear equations and admits well-known generalizations to Banach spaces and Riemannian manifolds. While we have opted for a reader-friendly presentation by placing most of the mathematical intricacies in the Appendix, these should not be overlooked, as they form the core of our contribution.
>
> **C)** Through our theoretical contributions, we also provide a framework for analyzing fair training by introducing several notions: the stereotypical gap, stereotypical predictors, fairness overcost, and a geometrical framework for understanding "stereotyped training," which can readily be adapted to other methods.
>
>
> We now proceed to answering point by point.
>
> **Reviewer 48E9:** "*Although the paper claims to be among the few theoretical treatments ... that directly address this issue from a theoretical standpoint [1,2,3].*"
>
>  **Authors:** We thank the referee for pointing us these recent papers. The reviewer is perfectly right on their relevance: we apologize. We will fix this issue carefully and mention these contributions in a section on related works. We will discuss several aspects of these works, as for instance, the quadratic cases and the phenomenon of Minority Initial Drop (MID) which clearly resonates with the majority-training zone effect.
>
> **Reviewer 48E9:** "*Key assumptions underlying the theoretical results ... under which they are likely to hold in practice.*"
>
>  **Authors:** We will clarify that point. At this stage we may say that when the proportions are considerably imbalanced, \$L\_0\$ becomes a perturbation of \$L\_1\$, and our assumptions are very likely to hold. Note also that we discussed the conditions in the linear regression case in Theorem 2 - a precised version of Theorem 1. In that case they have an appealing statistical nature as we see that  the majority/minority ratio of "variabilities" is also an important factor. Data with low variability are more likely to be "ignored" by standard training. This will be commented in the new version.
>
> **Reviewer 48E9:**  "*While the main theoretical insight is sound ... may not significantly advance the state of the art".*
>
>  **Authors:** We are unsure of how the statement of the reviewer should be understood, but for us the objective of this paper was to provide mathematical grounds to intuition and empirical observations of bias amplification. From that viewpoint we believe it is successful. Moreover, note also, that as mentioned in some of our other answers, we wish to emphasize that we also bring several novelties that allow for a real understanding of the issues at stake: metrics (gaps, ratios), geometrical insights, recommendation of longer training. See also general comments.
>
> **Reviewer 48E9:**  *Question: Please expand the related work on the theory of bias amplification, including how this work connects to and extends prior theoretical and empirical studies.*
>
>  **Authors:** We will definitely do this. In particular, we will add and comment the following content to the new related work section:
>
> [1] Francazi, E., Baity-Jesi, M. and Lucchi, A., 2023, July. A theoretical analysis of the learning dynamics under class imbalance. In International Conference on Machine Learning (pp. 10285-10322). PMLR.
>
> [2] Subramonian, A., Bell, S.J., Sagun, L. and Dohmatob, E., 2025. An Effective Theory of Bias Amplification. In International Conference on Learning Representations.
>
> [3] Mannelli, S.S., Gerace, F., Rostamzadeh, N. and Saglietti, L., Bias-inducing geometries: exactly solvable data model with fairness implications. In ICML 2024 Workshop on Geometry-grounded Representation Learning and Generative Modeling.
>
> [4] A systematic study of bias amplification, Hall, Melissa and van der Maaten, Laurens and Gustafson, Laura and Jones, Maxwell and Adcock, Aaron, arXiv preprint arXiv:2201.11706, 2022
>
> [5] An investigation of why overparameterization exacerbates spurious correlations, Sagawa, Shiori and Raghunathan, Aditi and Koh, Pang Wei and Liang, Percy, International Conference on Machine Learning, 8346--8356, 2020.
>
> [6] Kunster et al: "Heavy-Tailed Class Imbalance and Why Adam Outperforms Gradient Descent on Language Models" .
>
> **Reviewer 48E9:** *Question: Please include a brief description of the main assumptions (e.g., in Theorem 1) and discuss when they are expected to hold in practice.*
>
>  **Authors:** Yes, we will definitely do that. When the proportions are considerably imbalanced, \$L\_0\$ becomes a perturbation of \$L\_1\$, and our assumptions are very likely to hold. Other possibilities include differences in variability, as in the linear regression case, which have a more statistical nature (see the ratio after Theorem 2). See also our answer to Reviewer 4HDc.

---

> > ### Comment · Reviewer_48E9 · 2025-08-01
> >
> > Thank you for the detailed rebuttal. It addressed my questions and improved my understanding of the work. That said, it remains somewhat unclear how the contribution fits within the broader context of related work, and how realistic the main theoretical assumptions are. I look forward to seeing these aspects clarified and expanded in the revised version. Nonetheless, I find the proposed framework valuable for understanding bias amplification and currently lean toward acceptance.

---

> > > ### Author Response · Authors · 2025-08-01
> > >
> > > We thank the referee again for their helpful review. We will strive to present the literature fairly and to improve the clarity of the manuscript. We remain available for any further questions or precisions.

---

### Official Review · Reviewer_4k56 · 2025-07-03

**Clarity:** 3
**Significance:** 2
**Originality:** 3
**Rating:** 4
**Confidence:** 3

**Summary:**

This paper introduces a theoretical framework for analyzing majority–minority tasks. It shows the proximity between “full-data” and stereotypical predictors, analyzes the dominant region of the parameter space, and provides a lower bound on the additional budget required to “unlearn.” The authors also perform experiments to validate their theoretical results.

**Questions:**

Although the theoretical framework appears model-agnostic, the paper discusses only deep-learning models—why exclude other model classes?

Question on Line 99: Why does ∇L₁ = 0 imply ∇L = 0 even when L₁ is much larger than L₀?

**Ethical Concerns:**

["NO or VERY MINOR ethics concerns only"]

**Final Justification:**

The authors have replied to most of my concerns regarding the paper. My score remains unchanged as the reasons to accept outweigh the reasons to reject.

**Limitations:**

See weaknesses.

**Paper Formatting Concerns:**

NA.

**Quality:**

2

**Strengths And Weaknesses:**

**Strengths**

**Theoretical novelty**: The analysis of the distance between critical points in Theorem 1 is both novel and intuitive.

**Empirical validation and reproducibility**: The authors provide experiments on distances to critical points and include the linear regression case for clarity.

**Weaknesses**

**Unclear scope**: In the introduction, the authors situate this work in the bias literature, yet most experiments use an imbalanced-learning setup. It therefore remains unclear whether the results extend to learning from genuinely biased data.

**Restrictive binary setup**: The assumption of a binary majority/minority split is overly limiting, since real-world settings often involve intersectional minority groups.

**Missing test-set generalization**: It is surprising that test errors are not reported for the experiments presented in Table 1 and Figure 3.

---

> ### Author Rebuttal · Authors · 2025-07-30
>
> **General comment.**
>
>
> We greatly thank the referees for their useful comments, we believe that they will considerably improve the quality of the paper. We would like to make preliminary statements that will also impact the presentation of our revision.
>
> **A)** We will certainly include all the missing references —we apologize for the oversight, and we acknowledge their importance. A dedicated section on previous work will be added.
>
> **B)** Our contribution is principally theoretical and aims to explain the majority/minority scenario in the nonconvex setting. The empirical part is merely intended to illustrate our theorems, which is why it focuses solely on raw SGD. Our approach via the Kantorovich theorem is both non-trivial and notably robust to changes in assumptions, as it applies to nonlinear equations and admits well-known generalizations to Banach spaces and Riemannian manifolds. While we have opted for a reader-friendly presentation by placing most of the mathematical intricacies in the Appendix, these should not be overlooked, as they form the core of our contribution.
>
> **C)** Through our theoretical contributions, we also provide a framework for analyzing fair training by introducing several notions: the stereotypical gap, stereotypical predictors, fairness overcost, and a geometrical framework for understanding "stereotyped training," which can readily be adapted to other methods.
>
>
> We now proceed to answering point by point.
>
> **Reviewer 4k56:** "**Unclear scope**: *can the results extend to learning from genuinely biased data?*"
>
> **Authors:** Many thanks, we will indeed clarify this important point and emphasize it further in the introduction. Our goal is to explain how data imbalance leads to bias amplification in learned models. The theory is agnostic to why the imbalance arose; it requires only the imbalance conditions formalized in our theorems (i.e., a majority component and a minority component inducing asymmetric risk contributions as modeled by the ratio term after Theorem 2, that mixes the sampling effect and the variability effect). We shall provide a more direct and clear statement in the introduction, note however that the second paragraph came with a clear description of what could be the different notions of bias as well as the one we study in the paper.
>
> If we understand the referee correctly, "genuinely biased data" refers to bias introduced during the data collection or construction process, rather than bias arising from the learning algorithm itself. As such, it falls outside the scope of this paper.
>
>
> **Reviewer 4k56:**  **Restrictive binary setup**:  "*The assumption of a binary majority/minority split is overly limiting, since real-world settings often involve intersectional minority groups.*"
>
> **Authors:** The subject with two groups is not new, however the bias amplification phenomenon was not  explained in the nonconvex case. Along our theoretical perspectives, we believe it is the first natural and substantial step before moving to more complex ones. We also draw the reviewers's attention to the strong theoretical nature of the paper and to the substantial technical background required to understand this majority/minority scenario.  We also point out that our theory singled out several metrics/objects that were not used previously and that will be needed to understand more complex scenarios: stereotypical predictors, stereotype gap, "fair training=long training", fairness overcost ...
>
> However, we fully agree that intersectional minority groups are fundamental, and we hope to contribute to the development of a theory addressing this case in forthcoming papers.
>
> Please also refer to our response to Reviewer DEPM, which raises a similar concern.
>
> **Reviewer 4k56:** **Missing test-set generalization.**
>
> **Authors:** We greatly thank the reviewer for this remark. We will add test errors in the main paper and provide a Table 1 with test errors in Appendix C.
>
> **Reviewer 4k56:** *Question: "Although the theoretical framework appears model-agnostic, the paper discusses only deep-learning models—why exclude other model classes?"*
>
> **Authors:** While our fairness analysis is model-agnostic, deep learning was indeed our main target. Besides in DL, frameworks like PyTorch are well suited for tracking training dynamics (e.g., mini-batch updates, subgroup metrics). This is mainly why we focused on neural networks, including TabNet, a strong baseline for tabular data.
>
> Still, you will find below an XGBoost experiment on a logistic regression for Adult dataset. We train a single model incrementally: at each iteration, we add 10 more trees (using xgb_model=... to continue training from the previous booster), and evaluate performance on both train and test sets. We track global accuracy and subgroup accuracy (A = 0: minority, A = 1: majority) at regular time intervals.
>
> Training Time (s) | Train global | Train A=0 | Train A=1 | Test global | Test A=0 | Test A=1 |
> |-------------------|------------------|------------------|------------------|------------------|------------------|------------------|
> | 0.47 | 0.8772 | 0.5530 | 0.8893 | 0.8679 | 0.5263 | 0.8807 |
> | 7.70 | 0.8989 | 0.6734 | 0.9074 | 0.8679 | 0.5996 | 0.8781 |
> | 16.36 | 0.9098 | 0.7235 | 0.9168 | 0.8668 | 0.5940 | 0.8771 |
> | 26.45 | 0.9190 | 0.7623 | 0.9249 | 0.8649 | 0.5921 | 0.8752 |
> | 38.03 | 0.9261 | 0.7987 | 0.9308 | 0.8624 | 0.5977 | 0.8724 |
> | 51.11 | 0.9312 | 0.8205 | 0.9353 | 0.8606 | 0.5996 | 0.8705 |
> | 65.64 | 0.9359 | 0.8464 | 0.9393 | 0.8589 | 0.5959 | 0.8688 |
> | 82.94 | 0.9385 | 0.8529 | 0.9417 | 0.8579 | 0.5959 | 0.8678 |
>
> As in Deep Learning, we observe that longer training progressively improves the minority group’s accuracy, reducing the gap with the majority group. This effect is visible both in train and test performance. While the global test accuracy plateaus early, the test accuracy for A = 0 continues to increase with training time.
>
>
> **Reviewer 4k56:**  *"Question on Line 99: Why does $\nabla L_1 = 0$ imply $\nabla L = 0$ even when $L_1$ is much larger than $L_0$?"*
>
> **Authors:** We are confused as we did not write this implication on Line 99. We meant to informally refer to conclusion (i) of Theorem 1, that states that when there is a local minimizer for the majority loss $L_1$, then there is another local minimizer of the total loss L that is close in the parameter space. Of course, these two local minimizers are not exactly equal in general, so we did not mean to say that $\nabla L_1 (\theta) = 0$ exactly implies $\nabla L(\theta) = 0$.
> We will clarify this sentence, as it importantly relates to one of the core conclusions of the paper.

---

> > ### Comment · Reviewer_4k56 · 2025-08-06
> >
> > I thank the authors for their rebuttal, in particular the new experiments.
> >
> > I also thank the authors for their precision regarding local minimizers for L_1 and L.

---

> > > ### Author Response · Authors · 2025-08-06
> > >
> > > Thank you. We remain available for any further clarification.

---

### Official Review · Reviewer_DEPM · 2025-07-07

**Clarity:** 3
**Significance:** 2
**Originality:** 3
**Rating:** 4
**Confidence:** 3

**Summary:**

This paper investigates how gradient descent amplifies bias when training on imbalanced data with majority and minority groups. The authors develop a theoretical framework for majority-minority learning tasks where the total loss L = L1 + L0, with L0 (minority) being negligible compared to L1 (majority). They prove that critical points of the full loss L (representative predictors) are close to critical points of L1 (stereotypical predictors that ignore minority features), with distance bounded by a "stereotype gap" that depends on population and variance imbalance ratios. The analysis reveals that the parameter space is dominated by a "majority-training zone" where minimizing L essentially minimizes only L1. The authors derive lower bounds on the additional training time required to achieve fair predictions and validate their findings through experiments on CIFAR-10, EuroSAT, and Adult datasets.

**Questions:**

1. The paper uses "SGD with a constant learning rate... no weight decay or learning rate decay schedule was applied" to match the theoretical setting. How would the results change with other optimization practices (Adam, learning rate schedules, weight decay, pretrained models)? Would the bias amplification be worse or better?

2. Can the theoretical framework extend to settings with multiple minority groups?

**Ethical Concerns:**

["NO or VERY MINOR ethics concerns only"]

**Final Justification:**

The theoretical framework is solid, the experimental validation supports the claims, and the identified issues are addressable in revision. The core insight about the dominance of the majority-training zone and the quantification of fairness overcost provides valuable theoretical grounding for understanding bias amplification. The authors have satisfactorily addressed my concerns about the proof, committed to improving figure captions and adding a proper related work section. While the paper could benefit from more algorithmic solutions beyond ‘train longer’.

**Limitations:**

Yes

**Quality:**

2

**Strengths And Weaknesses:**

**Strengths:**
- The paper is well written and easy to understand.
- The concepts of "stereotype gap" and "majority-training zone" provide clear geometric intuition for why standard training amplifies bias in imbalanced settings.
- The lower bounds on debiasing time offer concrete guidance on how much additional training is needed to achieve fairness.
- The experiments are validated on different domains (vision and tabular datasets).

**Weaknesses:**

- Theorem Statement: For each pair of distinct elements θ,θ′∈crit L₁ (resp. crit L), we have ∥θ−θ′∥≥δ/32M.
However the Proof Issue: The proof derives two different bounds for critical points of L1 and L:

      - For crit L₁: ∥θ−θ′∥≥δ/2M

      - For crit L: ∥θ−θ′∥≥δ/32M
However, the theorem claims the same bound δ/(32M) for both. This contradicts the proof, which shows a stricter bound δ/(2M) for L1 and a looser bound δ/(32M) for L. This error affects the interpretation of critical point separation and needs correction.

- While the paper identifies the problem and quantifies training time, it doesn't propose algorithmic solutions beyond "train longer," missing opportunities for more efficient debiasing methods.

- The figures lacks consistency and clarity. Several figures (e.g., Figure 1 and Figures 3–5) are not accompanied by sufficiently descriptive captions, and the figure references in the main text are very short.  This makes it difficult for the reader to connect theoretical claims with empirical evidence. For example, the figure showing the majority and minority training zones (Figure 1) is central to the paper’s geometric intuition, but its caption is concise and does not adequately explain the experimental setup or relevance

- The paper lacks a dedicated related work section.

---

> ### Author Rebuttal · Authors · 2025-07-30
>
> **General comment.**
>
>
> We greatly thank the referees for their useful comments, we believe that they will considerably improve the quality of the paper. We would like to make preliminary statements that will also impact the presentation of our revision.
>
> **A)** We will certainly include all the missing references —we apologize for the oversight, and we acknowledge their importance. A dedicated section on previous work will be added.
>
> **B)** Our contribution is principally theoretical and aims to explain the majority/minority scenario in the nonconvex setting. The empirical part is merely intended to illustrate our theorems, which is why it focuses solely on raw SGD. Our approach via the Kantorovich theorem is both non-trivial and notably robust to changes in assumptions, as it applies to nonlinear equations and admits well-known generalizations to Banach spaces and Riemannian manifolds. While we have opted for a reader-friendly presentation by placing most of the mathematical intricacies in the Appendix, these should not be overlooked, as they form the core of our contribution.
>
> **C)** Through our theoretical contributions, we also provide a framework for analyzing fair training by introducing several notions: the stereotypical gap, stereotypical predictors, fairness overcost, and a geometrical framework for understanding "stereotyped training," which can readily be adapted to other methods.
>
>
> We now proceed to answering point by point.
>
>
>
> **Reviewer DEPM:** "*Theorem Statement: For each pair of distinct elements θ,θ′∈crit L₁ (resp. crit L), we have ∥θ−θ′∥≥δ/32M. However the Proof Issue: .... This error affects the interpretation of critical point separation and needs correction.*"
>
> **Authors:** We respectfully disagree and we think there might be a confusion of some sort. Indeed, there is no mathematical issue with the proof of Theorem 1. We have just chosen to provide only one lower bound (the smaller of the two) for concision in the theorem statement.
>
> (Details: As pointed out by the reviewer, the proof states
>
> - For crit L1, $|| \theta - \theta' || \ge \frac{\delta}{2M} $. Since $\frac{\delta}{2M} \ge \frac{\delta}{32M}$, we also have  $|| \theta - \theta' || \ge \frac{\delta}{32M} $
>
> - For crit L, $|| \theta - \theta' || \ge \frac{\delta}{32M} $.
>
> So it is indeed true that for each pair of distinct elements $\theta,\theta'$ in crit L1 (resp. crit L), we have $|| \theta - \theta' || \ge \frac{\delta}{32M} $. Note that it is  interesting to see in (ii) that the critical points of L are well-separated while the matching pairs of critical points of L and L1 are close. We shall explain this more clearly.)
>
>
> The referee's remark is nonetheless useful, as it highlights the need to place greater emphasis on point (i) of Theorem 1, which is the fundamental fact. To avoid confusion, we propose relocating point (ii) and underscoring point (i), which conveys the core result.
>
>
> **Reviewer DEPM:** "*While the paper identifies the problem and quantifies training time, it doesn't propose algorithmic solutions beyond "train longer," missing opportunities for more efficient debiasing methods."*
>
> **Authors:**  We have two answers to this:
>
> - The paper is theoretical: it highlights the key role of the Kantorovich theorem, the importance of second-order proximity, and, beyond the "train longer" recommendation, offers a flexible framework for measuring stereotype gaps both numerically and geometrically. We believe these properties and metrics are robust across different settings and provide a solid foundation for a general framework and further studies on other methods.
> Let us also mention, that while we agree that alternative methods may ultimately achieve greater fairness, our study makes us think that long training remains unavoidable when relying on first-order methods.
>
>
> - We shall also provide further references towards recent empirical/theoretical references that we missed and that will open perspectives towards other settings; [1,2,3] of Reviewer 48E9, and also Kunster et al: *"Heavy-Tailed Class Imbalance and Why Adam Outperforms Gradient Descent on Language Models"* .
>
> **Reviewer DEPM:** "*The figures lacks consistency and clarity ... but its caption is concise and does not adequately explain the experimental setup or relevance.*"
>
> **Authors:**  We totally agree and will improve the presentation. In particular, we will explain that within the majority training zone on Figure 1, learning is almost blind to L0, making little distinction between training with L1 and L. The learning of L0 is mainly activated outside or near the boundary of this zone.  Here is the addition to the caption that we propose: "The majority training zone, where only L1 is effectively learned, covers almost the entire space."
> For Figures 3 to 5, the referee is perfectly right again, we will clarify that prolonged training is fundamental to learn minority traits — and the smaller the minority, the longer the training required.
>
> **Reviewer DEPM:** "*The paper lacks a dedicated related work section.*"
>
> **Authors:** Definitely, we will add such a section including the references provided by Reviewer 48E9 and new ones that were communicated to us. We will also make more explicit our current references. We refer to the answer to Reviewer 48E9 for an example of a new content of this related work section.
>
> **Reviewer DEPM:** "*Question 1: The paper uses "SGD with a constant learning rate... no weight decay or learning rate decay schedule was applied" to match the theoretical setting. How would the results change with other optimization practices (Adam, learning rate schedules, weight decay, pretrained models)? Would the bias amplification be worse or better?*"
>
> **Authors:** This is indeed an important question that we plan to study into detail in the future. A comprehensive study of the many adjunction used in classical deep learning deserves to be understood independently, at least empirically, not to speak of the fact that some of them are probably extremely complex to study simultaneously.  In the spirit of the current paper, we could nevertheless include some precise references and a few new experiments to lay the groundwork for future work. For instance we ran the same experiments as Imbalanced CIFAR-10 with AdamW. In the table below we display the fairness overcost (in %) for $\kappa = 90%$ across imbalance levels  $\zeta \in$ [1%, 10%, 30%]. Means over 3 independent runs. Experiments were conducted on a single NVIDIA A100 80GB GPU with a total runtime of approximately 10 hours.
>
> | Model                               | Parameters | 1%  | 10% | 30% |
> |------------------------------------|------------|-----|-----|-----|
> | MobileNetV2  | 543K       | 326  | 310  | 214  |
> | VGG11 | 9M         | 401  | 244  | 169  |
> | ResNet18        | 11M        | 465  | 342  | 209  |
> | ResNet50                           | 25M        | 369  | 327  | 219  |
> | ResNet101                          | 42M        | 220  | 192  | 172  |
>
> Fairness overcost increases with imbalance, especially for smaller models. Larger models like ResNet101 are more robust, showing lower overcosts. This suggests that higher capacity helps reduce performance gaps under imbalance.
>
> Note that the overall recommendation "train longer" remains and proportions are similar to those of SGD.
>
>
>
> **Reviewer DEPM:** "*Question 2: Can the theoretical framework extend to settings with multiple minority groups?*"
>
> **Authors:** Our theory analyzes bias amplification under a binary partition of the data: a majority component and a single  minority subgroup. The loss is decomposed accordingly, and the proof techniques rely on this two‑group structure. The creation of the two subgroups is agnostic. As a result, any subgroup can be studied via a one versus the remaining part split, by repeating the analysis for each subgroup in turn (e.g., on CIFAR‑10, treating each class as the minority in separate runs after inducing imbalance).
> Yet the limitation is that  the current theory does not characterize joint interactions among multiple minority groups considered simultaneously. This is an interesting topic that will be the subject of a future work.

---

> > ### Comment · Reviewer_DEPM · 2025-08-05
> >
> > Thank you for the detailed rebuttal. I have carefully reviewed your responses. Upon re-reading, the proof is indeed correct and I appreciate your clarification. Given your thorough responses and commitments to revision, I will increase my score.

---

> > > ### Author Response · Authors · 2025-08-06
> > >
> > > Many thanks. We really appreciate this, and remain available if needed.

---

### Note · Authors · 2025-08-12

We take the opportunity of these “Author Final Remarks” to warmly thank the Area Chair and the four reviewers.

We appreciate the constructive comments from the reviewers, and their receptivity to our responses. These comments, and the subsequent discussions will help us to revise and improve the paper. Here, we list the main changes that we plan to make.

- We will revise the introduction, in particular clarifying the type of bias that is the focus of the paper.

- We will discuss the new references provided to us by the reviewers, and clarify our originality compared to them.

- We will better explain (and hierarchize) the meaning of the conclusions of our theoretical results. We will also better explain and justify the assumptions. In particular, we will explain that the assumptions of Theorem 1 hold simply when $L_1$ and $L_0$ are smooth Morse functions, that vanish outside of a fixed compact set, have at least one critical point in this set, with the Hessian of $L_0$ small enough.

- We will add the experimental results that we have given in our rebuttals to the paper, as we agree they tackle important points.

- The illustrative theoretical results from the quadratic case will be given in a single separate section, in order to focus more the exposition on the general nonconvex case.

- We will revise the figures' captions according to the reviewers' feedback.

---

### Decision · Program_Chairs · 2025-09-17

**Decision:**

Accept (poster)

**Comment:**

This paper studies bias amplification, and proposes a theoretical framework for majority-minority learning tasks, showing how standard training can favor majority groups and produce stereotypical predictors that neglect minority-specific features. The rebuttal successfully addressed most of the initial concerns (clarifications of the contributions, stronger empirical evidence, and related work analysis), leading to two reviewers raising their scores. After careful deliberation, the decision was made to accept this paper - congratulations to the authors! It is crucial, however, that all provided clarifications are incorporated in the final version of the paper.